# Aiolos represses CD4+ T cell cytotoxic programming via reciprocal regulation of TFH transcription factors and IL-2 sensitivity

Kaitlin A. Read[1,2,9], Devin M. Jones[1,2,9], Srijana Pokhrel[1], Emily D. S. Hales[1], Aditi Varkey[1], Jasmine A. Tuazon[1,2,3], Caprice D. Eisele[2,4], Omar Abdouni[1], Abbey Saadey[1,2], Melissa R. Leonard [1,5], Robert T. Warren[1], Michael D. Powell[6], Jeremy M. Boss[6], Emily A. Hemann[1,7], Jacob S. Yount[1,7], Gang Xin[1,8], Hazem E. Ghoneim [1,8], Chan-Wang J. Lio[1,8], Aharon G. Freud [4,8], Patrick L. Collins [1,8] & Kenneth J. Oestreich [1,7,8] ✉

During intracellular infection, T follicular helper (TFH) and T helper 1 (TH1) cells promote humoral and cell-mediated responses, respectively. Another subset, CD4-cytotoxic T lymphocytes (CD4-CTLs), eliminate infected cells via functions typically associated with CD8+ T cells. The mechanisms underlying differentiation of these populations are incompletely understood. Here, we identify the transcription factor Aiolos as a reciprocal regulator of TFH and CD4-CTL programming. We find that Aiolos deficiency results in downregulation of key TFH transcription factors, and consequently reduced TFH differentiation and antibody production, during influenza virus infection. Conversely, CD4-CTL programming is elevated, including enhanced Eomes and cytolytic molecule expression. We further demonstrate that Aiolos deficiency allows for enhanced IL-2 sensitivity and increased STAT5 association with CD4-CTL gene targets, including Eomes, effector molecules, and IL2Ra. Thus, our collective findings identify Aiolos as a pivotal regulator of CD4-CTL and TFH programming and highlight its potential as a target for manipulating CD4+ T cell responses.

Immune responses mounted against intracellular pathogens, such as influenza virus, require coordination between numerous immune cell types. These include the effector CD4+ T helper 1 (TH1), T follicular helper (TFH), and cytotoxic subsets, each of which perform distinct activities to eliminate infection. TH1 cells produce effector cytokines such as interferon gamma (IFN-γ) to both recruit and activate additional immune cell populations at sites of infection[1]. TFH populations are key participants in the generation of humoral immune responses through cognate B cell interactions, which support somatic hypermutation, class-switch recombination, affinity maturation, and the formation of long-lived plasma cell populations[2]. While CD4+ T cells have primarily been associated with the orchestration of

[1]Department of Microbial Infection and Immunity, The Ohio State University College of Medicine and Wexner Medical Center, Columbus, OH 43210, USA. [2]Biomedical Sciences Graduate Program, Columbus, OH 43210, USA. [3]Medical Scientist Training Program, Columbus, OH 43210, USA. [4]Department of Pathology, The Ohio State University College of Medicine and Wexner Medical Center, Columbus, OH 43210, USA. [5]Combined Anatomic Pathology Residency/PhD Program, The Ohio State University College of Veterinary Medicine, Columbus, USA. [6]Department of Microbiology and Immunology, Emory University School of Medicine, Atlanta, GA 30322, USA. [7]Infectious Diseases Institute, The Ohio State University College of Medicine and Wexner Medical Center, Columbus, OH 43210, USA. [8]Pelotonia Institute for Immuno-Oncology, The Ohio State Comprehensive Cancer Center, Columbus, OH 43210, USA. [9]These authors contributed equally: Kaitlin A. Read, Devin M. Jones. ✉e-mail: Ken.Oestreich@osumc.edu

immune responses, it is now widely accepted that they can also differentiate into CD4-cytotoxic T lymphocytes or 'CD4-CTLs'. CD4-CTLs are characterized by their ability to secrete both cytotoxic molecules and pro-inflammatory cytokines, ultimately resulting in the direct targeting and killing of virus-infected cells in a class two major histocompatibility complex (MHC-II)-restricted fashion[3,4]. In contrast to these protective roles, dysregulation of these subsets has been associated with autoimmune disorders in which inappropriate production of pro-inflammatory molecules and/or autoantibodies drive pathogenesis[4–6]. Thus, there has been interest in defining the mechanisms that govern the differentiation and function of the $T_H1$, $T_{FH}$, and CD4-CTL subsets during both healthy and dysregulated immune responses.

CD4+ T cell differentiation programs are guided to a large degree by cytokine-induced transcriptional networks[7]. For example, $T_{FH}$ cell differentiation requires signaling from STAT3 activating-cytokines, such as IL-6, which induce the expression and function of the transcriptional repressor B cell lymphoma 6 (Bcl-6)[8–11]. Bcl-6 supports $T_{FH}$ differentiation in part by inhibiting the expression of $T_{FH}$ antagonists, such as Blimp-1, as well as additional genes associated with alternative cells fates, including those important for $T_H1$, CD4-CTL, $T_H2$, and $T_H17$ differentiation[8,9,11–13]. While Bcl-6 is considered the lineage-defining transcription factor for $T_{FH}$ development, $T_{FH}$ differentiation also relies upon TCF-1 (encoded by the gene *Tcf7*), Lef-1, Tox, Tox2, Maf, Batf, Ascl2, and Irf4[2,14–18]. Of these, Tox2, Batf, and TCF-1 have been implicated in the direct induction of *Bcl6* expression[5,19,20]. Like Bcl-6, TCF-1 also represses the expression of Blimp-1, and has been implicated in the negative regulation of both $T_H1$ and CD4-CTL differentiation[17,20–22]. Finally, the transcription factor Zfp831 was recently identified as a positive regulator of both *Tcf7* and *Bcl6*, and thus may play a prominent role in initiating $T_{FH}$ cell differentiation[23]. Together, these early factors both set the stage for $T_{FH}$ development and function to oppose alternative gene programs.

With regard to $T_H1$ and CD4-CTL differentiation, a second cytokine pathway, IL-12/STAT4, promotes expression of the T-box transcription factor T-bet, which is required for $T_H1$ differentiaiton and the production of IFN-γ[24–27]. T-bet has also been implicated in regulation of cytotoxic programming via its ability to transactivate granzyme B expression[4,28]. $T_H1$ and CD4-CTL differentiation further require the transcription factor Blimp-1, which represses expression of alternative polarization programs including those underlying the $T_{FH}$ subset[8,13]. While the $T_H1$ and CD4-CTL subsets share some regulatory features, including a dependence on T-bet and Blimp-1, there are also differences in their respective transcriptional regulators[4,29]. These include a second T-box transcription factor, Eomesodermin (Eomes), and Runx3, which have been associated predominantly with cytotoxic programming[3,30,31].

In addition to the above pathways, IL-2/STAT5 signaling has also emerged as a key determinant of effector subset programming, as it supports $T_H1$ and CD4-CTL differentiation, while opposing the $T_{FH}$ gene program[13,32–36]. In $T_H1$ and CD4-CTL populations, STAT5 directly induces the expression of Blimp-1. STAT5 further competes with, and antagonizes the activity of, STAT3, which allows for the repression of STAT3-induced genes including *Bcl6*[13]. Conversely, IL-6 and IL-21-activated STAT3 can repress STAT5-dependent induction of IL-2 receptor expression through an analogous competitive mechanism, thereby reducing STAT5 signaling[37]. Thus, the available literature suggests that opposing networks of environmental and transcriptional regulators function to maintain the appropriate balance of $T_{FH}$, $T_H1$, and CD4-CTL effector responses during infection. However, the comprehensive identity of the factors that comprise these networks, as well as the mechanisms these factors employ to govern effector subset programming, remain incompletely understood.

Here, we identify the Ikaros zinc finger (IkZF) transcription factor Aiolos as a pivotal regulator of the $T_{FH}$ and CD4-CTL differentiation programs. We find that Aiolos potentiates $T_{FH}$ differentiation, at least in part, through direct induction of the $T_{FH}$ transcriptional regulators Bcl-6 and Zfp831. In agreement with these findings, Aiolos deficiency compromises $T_{FH}$ differentiation and allows for induction of a cytotoxic-like program. Mechanistically, we find that Aiolos-deficient cells express elevated levels of IL-2R subunits, which manifests in augmented sensitivity to IL-2 signaling and increased STAT5 association with key CD4-CTL gene targets. These targets include CTL-associated transcription factors (Eomes, Blimp-1), effector molecules (granzyme B, perforin, IFN-γ), and IL-2Rα. Consequently, Aiolos-deficient CD4+ T cells exhibit increased production of cytotoxic effector molecules, including perforin and granzyme B, during responses to influenza infection. Together, these findings identify Aiolos as a reciprocal regulator of $T_{FH}$ and CD4-CTL differentiation and support its potential as a target for the therapeutic manipulation of CD4+ T cell-dependent humoral and cytotoxic responses.

## Results

### Aiolos expression is elevated in $T_{FH}$ cell populations

In a screen of in vitro-generated CD4+ T effector subsets, we found that Aiolos expression was elevated in in vitro-generated $T_{FH}$-like populations relative to $T_H1$ and $T_H2$ cells (Supplementary Fig. 1a and ref. [38]). Previous work established that Aiolos is also expressed in $T_H17$ populations[39]. We compared Aiolos expression between $T_{FH}$-like and $T_H17$ cells, and found a trending, but not significant, increase in Aiolos transcript in $T_H17$ cells relative to $T_{FH}$ (Supplementary Fig. 1b). To examine the kinetics of Aiolos expression, we compared naïve CD4+ T cells to $T_{FH}$-like and non-polarized ($T_H0$) cells. Aiolos transcript was elevated in $T_{FH}$-like cells relative to both naïve and $T_H0$ controls at 24 h, and correlated with that of *Bcl6* even at this early timepoint (Supplementary Fig. 1c, d). Collectively, these data suggested that Aiolos may positively regulate $T_{FH}$ programming.

We next sought to determine the role of Aiolos in the formation and function of $T_{FH}$ cells generated in vivo. We began by infecting wildtype (WT) C57BL/6J mice with influenza A/PR/8/34 (H1N1) virus, commonly termed 'PR8', and examined Aiolos expression in naive CD4+ T cells, PD-1+Cxcr5+ $T_{FH}$ cells, and non-$T_{FH}$ populations in lung-draining lymph nodes (DLN) (Fig. 1a and Supplementary Fig. 2a). Consistent with a role for Aiolos in $T_{FH}$ populations generated in vivo, Aiolos protein expression was significantly elevated in $T_{FH}$ populations relative to naïve CD4+ and non-$T_{FH}$ effector cells. Further, we found that Aiolos was highest in germinal center (GC) $T_{FH}$ cells, characterized by high expression of PD-1 and Cxcr5 (Fig. 1a). To determine whether Aiolos is similarly increased in human $T_{FH}$ cell populations, we analyzed previously published RNA-seq data for CD4+ naïve, non-$T_{FH}$ effector, and $T_{FH}$ cell populations from human tonsils (GSE58597)[40]. Indeed, human Aiolos (*IKZF3*) was most highly expressed in $T_{FH}$ populations, and its expression correlated with that of key $T_{FH}$ transcriptional regulators (*BCL6, TOX, TOX2*) and cell surface receptors (*CD40LG, PDCD1*, and *IL6R*) (Supplementary Fig. 2b, c). Expression of genes associated with antagonizing the $T_{FH}$ gene program, including *PRDM1* (which encodes BLIMP-1) and *IL2RA*, inversely correlated with Aiolos expression (Supplementary Fig. 2c). We next examined Aiolos protein expression in human tonsillar $T_{FH}$ and non-$T_{FH}$ effector cells. As with transcript, Aiolos protein expression was elevated in $T_{FH}$ populations relative to non-$T_{FH}$ effector cells (Supplementary Fig. 2d, e). Together, these findings supported a conserved role for Aiolos as a potential positive regulator of $T_{FH}$ differentiation in both mice and humans.

### Aiolos-deficient mice display defects in $T_{FH}$ differentiation

To determine whether Aiolos was required for the generation of $T_{FH}$ populations in vivo, we infected Aiolos-deficient (*Ikzf3*−/−; Supplementary Fig. 3a, b) or WT mice with PR8 and assessed influenza nucleoprotein (NP311)-specific $T_{FH}$ populations residing in the DLN via flow

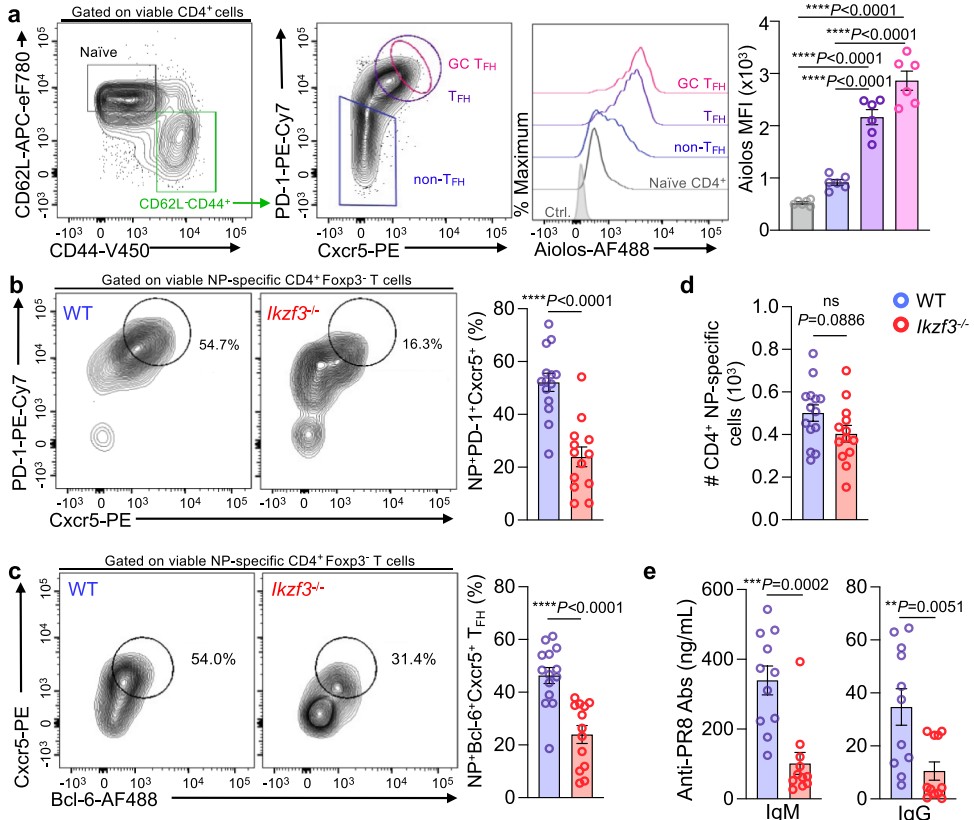

**Fig. 1 | Aiolos expression is elevated in $T_{FH}$ cell populations, and its loss results in disrupted $T_{FH}$ cell differentiation and antibody production in response to influenza infection. a** Naïve WT C57BL/6 mice were infected intranasally with 30 PFU influenza (A/PR8/34; "PR8") for 8 days. Single-cell suspensions were generated from the lung-draining lymph nodes (DLN), and analysis of Aiolos protein expression in the indicated populations was performed via flow cytometry. Data are compiled from 2 independent experiments ($n = 6 \pm$ s.e.m; ****$P < 0.0001$; one-way ANOVA with Tukey's multiple comparison test). **b–d** Analysis of the percentage of influenza nucleoprotein (NP)-specific PD-1$^{HI}$Cxcr5$^{HI}$ ($T_{FH}$) populations generated in response to influenza infection. Following single-cell suspension, cells were stained with fluorochrome-labeled tetramers to identify NP-specific populations. **b, c** Analysis of the percentage of influenza nucleoprotein (NP)-specific Bcl-

6$^{HI}$Cxcr5$^{HI}$ ($T_{FH}$) populations generated in response to influenza infection (For 'b', $n = 14$ for WT and 13 for Aiolos KO. For 'c', $n = 14$. Data are presented as mean ± s.e.m; data are compiled from four independent experiments; ****$P < 0.0001$; two-sided, unpaired Student's $t$ test). **d** Total NP-specific CD4$^+$ T cells generated in WT versus Aiolos-deficient animals following influenza infection were enumerated. ($n = 14$ for WT and 13 for Aiolos KO. Data are presented as mean ± s.e.m; data are compiled from 4 independent experiments; ****$P < 0.0001$; two-sided, unpaired Student's $t$ test. **e** ELISA analysis of indicated serum antibody levels in ng/mL at 8 d.p.i. Data are compiled from three independent experiments ($n = 11 \pm$ s.e.m, **$P < 0.01$, ***$P < 0.001$; two-sided, unpaired Student's $t$ test). Source data are provided as a Source Data file.

cytometry (Supplementary Fig. 3c). Consistent with our hypothesis, $Ikzf3^{-/-}$ mice exhibited a significant decrease in the percentage of antigen-specific $T_{FH}$ cells (identified as PD-1$^+$Cxcr5$^+$ or Bcl-6$^+$Cxcr5$^+$) relative to WT controls (Fig. 1b, c). This reduction was not due to an overall difference in NP-specific cells in the DLN of Aiolos-deficient mice, as these numbers were relatively unchanged between genotypes (Fig. 1d). We also observed a significant reduction in both the level of Bcl-6 protein expression and frequency of NP-specific Bcl-6$^+$ cells in Aiolos-deficient CD4$^+$ T cells at an early timepoint post-infection, supporting a role for Aiolos in the positive regulation of $Bcl6$ expression in vivo (Supplementary Fig. 3d). Again, this was not a result of altered NP-specific CD4$^+$ T cell numbers in the absence of Aiolos (Supplementary Fig. 3e). To determine whether the defect in $T_{FH}$ cell populations resulted in a functional impact on humoral responses, we measured influenza-specific antibody production via serum ELISA. We observed a significant reduction in flu-specific serum IgM and IgG in $Ikzf3^{-/-}$ animals relative to WT controls (Fig. 1e).

We considered the possibility that loss of Aiolos may result in alterations to regulatory CD4$^+$ T cell populations, as this could impact both $T_{FH}$ differentiation and influenza-specific antibody production. Indeed, in the absence of Aiolos, we observed a significant increase in the percentage of both bulk and antigen-specific activated

CD4$^+$CD44$^+$Foxp3$^+$ $T_{REG}$ cells, while Foxp3$^+$Cxcr5$^+$ $T_{FR}$ cells exhibited a slight but significant reduction in frequency (Supplementary Fig. 3f–h). Thus, these data suggest that Aiolos is required to fine-tune the balance of CD4$^+$ T cells in both $T_{FH}$ and regulatory CD4$^+$ T cell compartments.

## The $T_{FH}$ transcriptional program is disrupted in the absence of Aiolos

As our in vivo findings suggested that Aiolos may be required to regulate $T_{FH}$ responses during influenza virus infection, we wanted to determine whether there was a CD4$^+$ T cell-intrinsic role for Aiolos in regulating the $T_{FH}$ cell differentiation program. To examine alterations in the $T_{FH}$ transcriptome in the absence of Aiolos, we cultured naïve WT or Aiolos-deficient CD4$^+$ T cells under $T_{FH}$-polarizing conditions and performed RNA-seq analysis. PCA analysis of DESeq2-normalized reads revealed that WT vs. Aiolos-deficient samples formed distinct clusters, with genotype explaining over 79% of the variance (Supplementary Fig. 4a). Hierarchical clustering analysis revealed that Aiolos-deficient cells exhibited a distinct gene expression profile (Supplementary Fig. 4b). Genes downregulated in the absence of Aiolos included those encoding key pro-$T_{FH}$ transcriptional regulators ($Bcl6$, $Tox$, $Tox2$, $Tcf7$, and $Zfp831$) and additional $T_{FH}$-associated genes

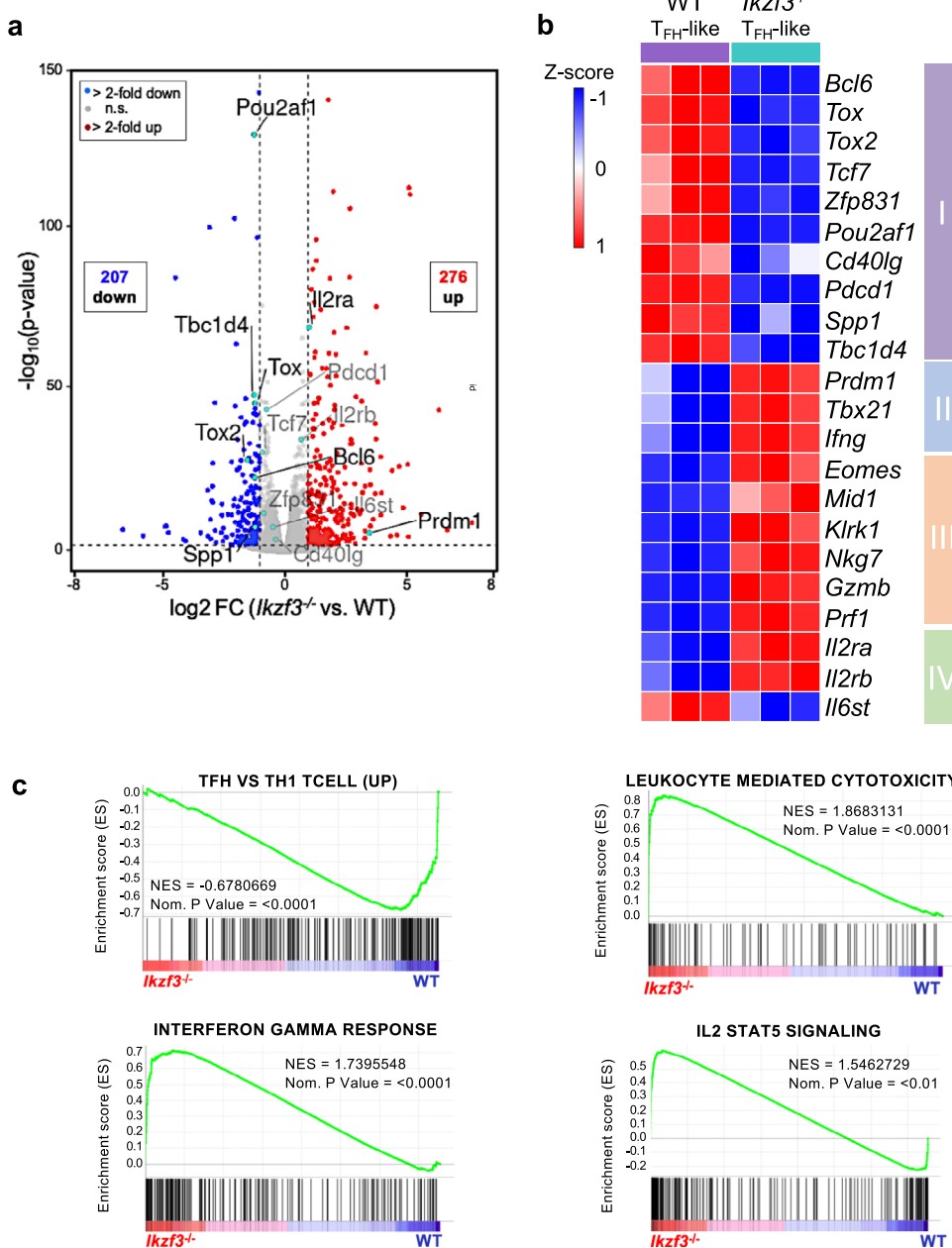

**Fig. 2 | Loss of Aiolos results in alterations in the T$_{FH}$ transcriptome.** Naïve WT or Aiolos-deficient (*Ikzf3$^{-/-}$*) CD4$^+$ T cells were cultured under T$_{FH}$-polarizing conditions for 3 days. RNA-seq analysis was performed to assess differentially expressed genes (DEGs) between WT and Aiolos-deficient cells. **a** Volcano plot displaying gene expression changes in Aiolos-deficient vs WT cells; genes of particular interest are labeled. Genes were color-coded as follows: no significant changes in expression (gray), upregulated genes with >2-fold change in expression with a $P < 0.05$ (red), downregulated genes with >2-fold change in expression with a $P < 0.05$ (blue) ($n = 3$; two-sided DESeq2 analysis). **b** Representative heat-maps of DEGs positively and negatively associated with the T$_{FH}$ gene program in WT vs. Aiolos-deficient cells; changes in gene expression are presented as row (gene) Z-score. **c** Pre-ranked (sign of fold change × −log$_{10}$(*p*-value)) genes were analyzed using the Broad Institute Gene Set Enrichment Analysis (GSEA) software for comparison against 'hallmark', 'gene ontology', and 'immunological signature' gene sets. Enrichment plots for indicated gene sets are shown. Data are compiled from three biological replicates from three independent experiments ($n = 3$; pre-ranked GSEA analysis). Source data are provided as a Source Data file.

including *Cd40lg*, which is critical for T$_{FH}$-B cell interactions (Fig. 2a, b box I). As Tcf-1 (encoded by *Tcf7*) has been implicated in regulating early T cell development, we wanted to determine whether its expression was similarly reduced in Aiolos-deficient naïve CD4$^+$ T cells[41,42]. However, analysis of *Tcf7* expression in naive Aiolos-deficient CD4$^+$ T cells revealed no loss of expression (Supplementary Fig. 4c).

In contrast to reduced expression of numerous T$_{FH}$ genes, expression of *Prdm1*, the gene encoding the T$_{FH}$ antagonist Blimp-1,

was markedly increased in the absence of Aiolos (Fig. 2a, b box II). Blimp-1 has also been linked to the differentiation of T$_H$1 and CD4-CTL populations. Strikingly, we observed marked increases in the expression of additional genes associated with the T$_H$1 (*Tbx21*, *Ifng*) and CD4-CTL (*Tbx21*, *Eomes*, *Ifng*, *Prf1*, *Gzmb*) subsets in the absence of Aiolos (Fig. 2b, box II, III). We further observed an increase in the expression of genes encoding the IL-2 receptor subunits IL-2Rα (CD25) and IL-2Rβ (CD122) in Aiolos-deficient cells, while *Il6st* (encoding the IL-6R subunit gp130) was decreased (Fig. 2a, b box IV). This was notable, as IL-2/

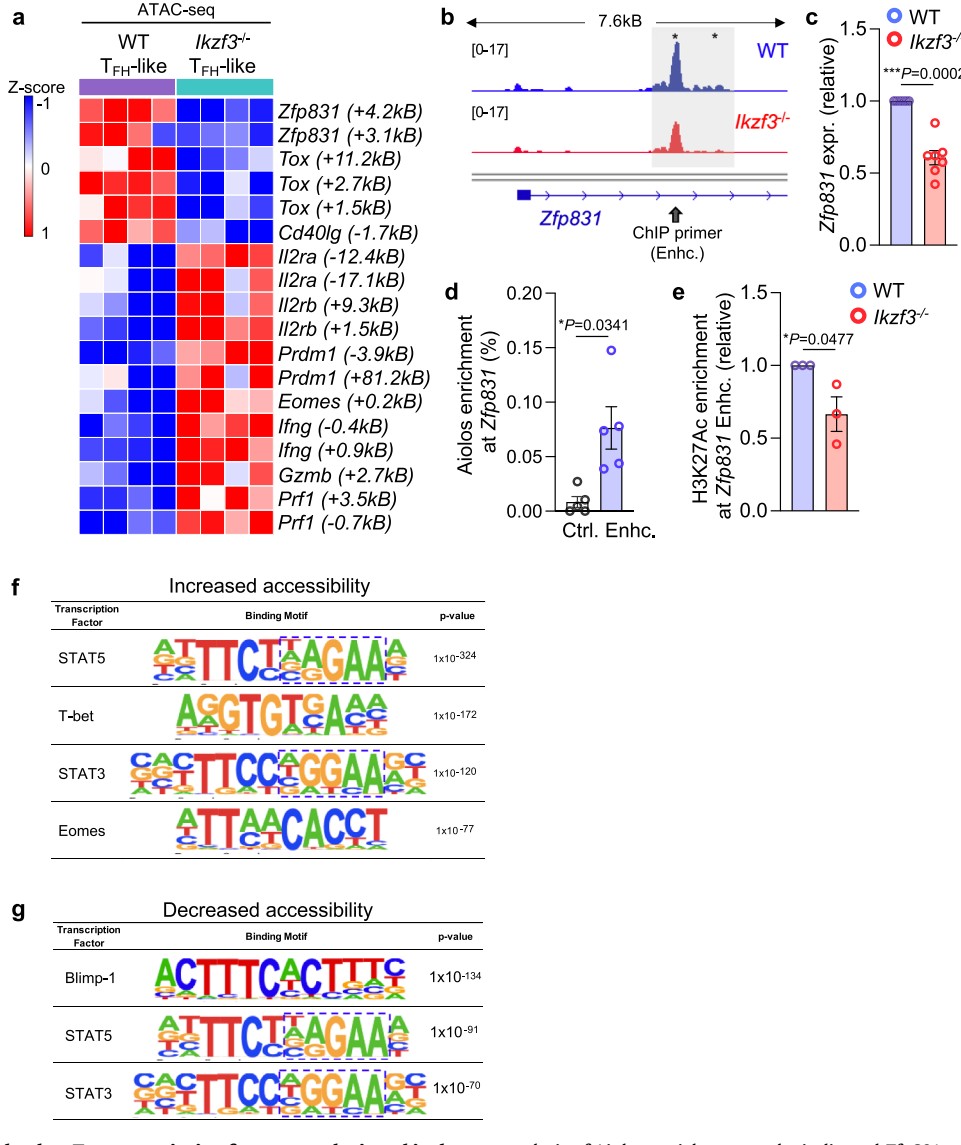

**Fig. 3 | Aiolos is enriched at key T_FH transcription factor gene loci, and its loss results in altered chromatin accessibility.** Naïve CD4+ T cells were cultured under T_FH-like polarizing conditions for 3 days. **a** Representative heatmap showing normalized counts from Assay for Transposase-Accessible Chromatin (ATAC)-seq of WT and Aiolos-deficient CD4+ cells. Loci shown exhibit statistically significant differences in accessibility between WT and Aiolos-deficient samples. Data are compiled from 4 independent experiments. **b** CPM-normalized representative IGV tracks for the *Zfp831* locus from four independent experiments are shown. Approximate location of primers used for chromatin immunoprecipitation (ChIP) is indicated with gray arrows. **c** Transcript analysis for *Zfp831* was performed via qRT-PCR. Data were normalized to *Rps18* control and are presented relative to the WT sample ($n = 7 \pm$ s.e.m; **$P < 0.01$, two-sided, unpaired Student's *t* test). **d** ChIP

analysis of Aiolos enrichment at the indicated *Zfp831* regulatory region or negative control region in T_FH-like cells are shown. Data were normalized to total input and are presented as percent enrichment of total minus IgG. ($n = 5 \pm$ s.e.m; *$P < 0.05$, **$P < 0.01$; two-sided, paired Student's *t* test). **e** ChIP analysis of H3K27Ac at the site of Aiolos enrichment in WT versus *Ikzf3−/−* cells. Data were normalized to total input, and IgG background was subtracted from each sample. Data are presented relative to the WT sample ($n = 5 \pm$ s.e.m; *$P < 0.05$, **$P < 0.01$; two-sided, unpaired Student's *t* test). **f, g** HOMER motif enrichment analysis of regions of increased (**f**) and decreased (**g**) accessibility in Aiolos-deficient samples. Data are compiled from analysis of four independent ATAC-seq experiments ($n = 4$, HOMER hypergeometric analysis). Source data are provided as a Source Data file.

STAT5 signaling is known to not only antagonize T_FH differentiation, but also induce the expression of numerous genes associated with the T_H1 and CD4-CTL gene programs[4,34]. Conversely, IL-6/STAT3 signaling is known to promote T_FH gene expression, yet repress IL-2 responsiveness[37]. Consistent with these findings, gene set enrichment analysis (GSEA) of hallmark and immunologic signature gene sets revealed that genes upregulated in T_FH populations were among the most downregulated in the absence of Aiolos, while gene sets associated with interferon gamma signaling, cytotoxicity, and IL-2-STAT5 signaling were significantly elevated in Aiolos-deficient samples (Fig. 2c, Supplementary Fig. 4d). Together, these data suggest that Aiolos may not only promote T_FH differentiation through activation of

key transcriptional regulators, but also safeguard the T_FH gene program by repressing genes associated with alternative (T_H1, CD4-CTL) subsets, including those important for IL-2/STAT5 signaling.

## Aiolos associates with and alters chromatin structure at T_FH gene regulatory elements

RNA-seq and confirmatory qRT-PCR analyses revealed reduced expression of key T_FH genes in Aiolos-deficient cells cultured under T_FH-like polarizing conditions (Fig. 2a, b, Fig. 3c, Supplementary Fig. 6b). As Ikaros zinc finger family members are known to remodel chromatin structure, we performed Assay for Transposase-Accessible Chromatin (ATAC)-seq analyses using WT versus *Ikzf3−/−* T_FH-polarized

cells to determine whether observed gene expression changes correlated with alterations to the chromatin landscape (Supplementary Fig. 5)[43]. We observed broad changes in chromatin accessibility in the absence of Aiolos, including statistically significant reductions in accessibility at numerous key $T_{FH}$ gene loci (Fig. 3a, Supplementary Figs. 5b, 6a). Notably, this included *Zfp831*, which was recently identified as a direct inducer of both Bcl-6 and Tcf-1 (*Tcf7*), as well as *Tox*, another key $T_{FH}$ transcriptional regulator, and *Cd40lg* (Fig. 3a, b, Supplementary Fig. 6a)[14,16–18,22,23]. In contrast, numerous genes associated with both $T_H1$ and CD4-CTL programs exhibited significant increases in accessibility in the absence of Aiolos, including *Prdm1*, *Eomes*, *Ifng*, *Gzmb*, and *Prf1* (Fig. 3a). Consistent with our transcript analysis indicating enhanced IL-2R subunit expression, we also observed numerous sites of significantly enhanced accessibility at both the *Il2ra* and *Il2rb* loci in Aiolos-deficient cells (Fig. 3a).

Given the significant reduction in both transcript expression and chromatin accessibility at the *Zfp831* locus in the absence of Aiolos, we next wanted to determine whether Aiolos may directly induce its expression (Fig. 3b, c). ChIP analyses revealed that Aiolos was significantly enriched at the differentially accessible intronic enhancer element for *Zfp831*, relative to a control region which did not exhibit altered accessibility (Fig. 3d). Further, we observed reduced enrichment of the chromatin mark H3K27Ac, which is associated with active enhancers, at this site in the absence of Aiolos (Fig. 3e). Together, these data suggest that Aiolos functions to directly induce expression of *Zfp831*, and that its loss results in reduced chromatin accessibility and enhancer activity at this region.

Our prior study implicated Aiolos in the direct regulation of *Bcl6*[38]. Consistent with these findings, we observed reduced Bcl-6 transcript and protein expression in Aiolos-deficient cells (Fig. 2b, Supplementary Fig. 1d, 6b). Further, at the site of Aiolos enrichment within the *Bcl6* promoter, we observed both reduced accessibility and a loss of H3K27Ac in the absence of Aiolos (Supplementary Fig. 6c–e). Together, these data suggest that Aiolos functions early during $T_{FH}$ differentiation to induce key transcriptional regulators.

We next performed motif analyses at regions of significantly increased and decreased accessibility in the absence of Aiolos (Fig. 3f, g). Consistent with increased transcript expression of $T_H1$- and CD4-CTL-associated genes, we observed enrichment of motifs for the transcriptional activators T-bet and Eomes at sites with increased accessibility (Fig. 3f). Conversely, sites with reduced accessibility were enriched for motifs for the transcriptional repressor Blimp-1 (Fig. 3g). These findings were consistent with observed upregulation of *Tbx21* (T-bet), *Eomes*, and *Prdm1* (Blimp-1) in the absence of Aiolos (Fig. 2a, b). Further, STAT5 and STAT3 binding motifs were enhanced at regions of altered chromatin accessibility (Fig. 3f, g). Notably, the core Ikaros family member binding sequence (GGGAA) was represented in these motifs, suggestive of a mechanistic interplay between Aiolos and STAT factors (Fig. 3f, g, dashed boxes). These findings were consistent with previous studies suggesting that Ikaros engages in antagonism with STAT5 in lymphoid populations, and suggested that Aiolos may engage in a similar mechanism to regulate effector CD4⁺ T subset programming[44,45].

### Aiolos deficiency potentiates CD4-CTL programming

RNA-seq analyses of Aiolos-deficient cells cultured under $T_{FH}$-like polarizing conditions revealed not only compromised $T_{FH}$ cell programming, but also augmented expression of key $T_H1$ and CD4-CTL genes (Fig. 2a, b). These data are consistent with studies demonstrating that regulators of $T_{FH}$ differentiation, including Bcl-6 and TCF-1, repress aspects of both $T_H1$ and CD4-CTL programs, which share many regulatory features[8,9,11,21]. Thus, we hypothesized that Aiolos may function to repress $T_H1$ and CD4-CTL programming. We therefore sought to examine the role of Aiolos in regulating these populations by culturing naïve WT and Aiolos-deficient CD4⁺ T cells

under $T_H1$-polarizing conditions. We first assessed IFN-γ, an effector cytokine produced by both $T_H1$ and CD4-CTLs, and found that it was elevated at the transcript and protein levels in the absence of Aiolos (Fig. 4a, d). However, transcript of the $T_H1$ lineage-defining transcription factor T-bet (encoded by *Tbx21*) was unchanged in the absence of Aiolos, suggesting that Aiolos may not repress $T_H1$ differentiation in this context (Fig. 4a). Rather, expression of the related T-box factor Eomes, a known positive regulator of the cytotoxic programs of CD4⁺ and CD8⁺ T cells, was significantly increased in Aiolos-deficient cells, suggesting that Aiolos may function to repress CD4-CTL programming (Fig. 4a, Supplementary Fig. 7a)[4]. As with $T_{FH}$ cell populations, we also observed significantly reduced *Bcl6* expression in the absence of Aiolos, suggesting that loss of this repressor may contribute to the observed phenotype (Fig. 4a). Relative to $T_H1$ cells, CD4-CTL populations produce increased IFN-γ, as well as cytolytic molecules including granzymes and perforin[4]. To determine whether Aiolos-deficient cells also exhibited enhanced cytotoxic effector molecule production, we assessed granzyme B and perforin expression in wildtype versus Aiolos-deficient CD4⁺ T cells cultured under $T_H1$-polarizing conditions. Indeed, Aiolos-deficient cells exhibited increased *Gzmb* and *Prf1* transcript expression (Fig. 4b, c) and produced significantly more granzyme B and perforin protein upon stimulation (Fig. 4e, f).

To gain further insight into the role of Aiolos in regulating CD4-CTL programming, we performed RNA-seq analysis of WT and Aiolos-deficient cells cultured under $T_H1$ conditions (Fig. 4g, Supplementary Fig. 7b–d). As in $T_{FH}$-like cells, we observed broad alterations to the transcriptome in the absence of Aiolos, including increased expression of transcription factors (*Prdm1*, *Runx3*), cell surface receptors (*Klrk1*, *Klrc1*), effector molecules (*Ifng*, *Gzmb*, *Prf1*), and additional genes (*Mid1*, *Nkg7*) associated with the CD4-CTL gene program (Fig. 4g, Supplementary Fig. 7c, d). In contrast, transcription factors associated with $T_{FH}$ differentiation and antagonists of CTL-differentiation (*Bcl6*, *Tcf7*, and *Zfp831*) were downregulated in the absence of Aiolos (Fig. 4g, Supplementary Fig. 7d). Consistent with these findings, GSEA revealed that gene sets associated with cell killing were among the most upregulated in the absence of Aiolos (Fig. 4h). Collectively, these data indicate that Aiolos plays a role in repressing the CD4-CTL gene program.

As in $T_{FH}$ cells, we also found that loss of Aiolos resulted in increased expression of *Il2ra* and *Il2rb* in $T_H1$-polarized populations (Fig. 4g). GSEA analyses again identified gene sets associated with IL-2/STAT5 signaling as among the most upregulated sets in Aiolos-deficient cells cultured under $T_H1$-polarizing conditions (Fig. 4i). As *Il2ra* expression has been linked to T cell activation, we sought to determine whether loss of Aiolos resulted in broad alterations to CD4⁺ T cell activation markers. RNA-seq analysis of both $T_H1$ and $T_{FH}$-like populations revealed that gene expression changes were not wholly conserved between the cell types, and that activation markers were not globally upregulated at the transcript level in the absence of Aiolos in polarized populations (Supplementary Fig. 8).

### Aiolos deficiency results in acquisition of cytotoxic features by CD4 T cells in vivo

During our studies, we found that Aiolos-deficient animals lost signifcantly less weight than their WT counterparts in response to influenza virus infection (Supplementary Fig. 9a). To determine whether this may be due in part to augmented $T_H1$ or CD4-CTL responses, we first assessed T-bet and Eomes expression in influenza nucleoprotein (NP)-specific CD4⁺ T cells in the DLN and lung of infected mice via flow cytometry. Consistent with our in vitro findings, we did not observe an increase in the percentage of T-bet⁺ NP-specific cells from the DLN or lungs of WT and *Ikzf3⁻/⁻* mice (Fig. 5a, b, Supplementary Fig. 9b). However, the percentage of Eomes⁺ NP-specific cells was enhanced in both the DLN and lungs of Aiolos-

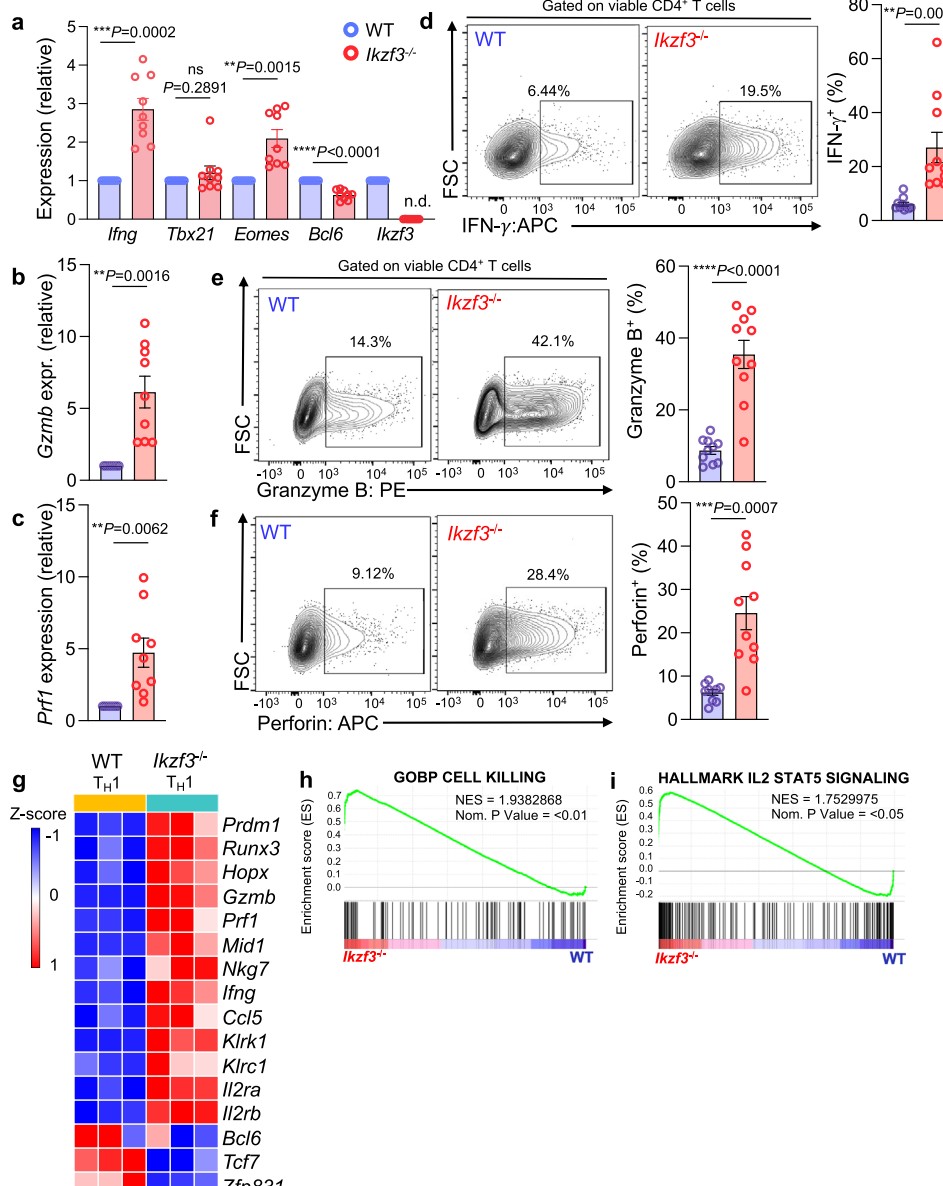

**Fig. 4 | Aiolos deficiency promotes CD4-CTL, but not $T_H1$ transcription factors and molecules.** Naive WT or Aiolos-deficient ($Ikzf3^{-/-}$) CD4$^+$ T cells were cultured under $T_H1$-polarizing conditions for 3 days. **a–c** Transcript analysis for the indicated genes was performed via qRT-PCR. Data are normalized to $Rps18$ control and presented relative to the WT sample for each gene. Data are compiled from 6 independent experiments. ($n = 9 \pm$ s.e.m; **$P < 0.01$, ***$P < 0.00$; two-sided, unpaired Student's $t$ test). **d–f** Analysis of cytotoxic effector molecule production by flow cytometry. Cells were cultured on stimulation with protein transport inhibitors for 3 h prior to analysis. ($n = 10 \pm$ s.e.m; **$P < 0.01$, ***$P < 0.001$, ****$P < 0.0001$; two-sided, unpaired Student's $t$ test). **g–i** RNA-seq analysis was performed to assess differentially expressed genes (DEGs) between WT and Aiolos-deficient cells. **g** Representative heatmap of DEGs positively and negatively associated with the $T_H1$, CD4-CTL, and $T_{FH}$ gene programs in WT vs. Aiolos-deficient cells is shown; changes in gene expression are presented as row (gene) Z-score. Data are compiled from three independent experiments. **h–i** Pre-ranked (sign of fold change × $-\log_{10}(p\text{-value})$) genes were analyzed using the Broad Institute Gene Set Enrichment Analysis (GSEA) software for comparison against 'hallmark' and 'immunological signature' gene sets. Enrichment plots for indicated gene sets are shown. Data are compiled from three biological replicates from three independent experiments ($n = 3$; pre-ranked GSEA analysis). Source data are provided as a Source Data file.

deficient mice compared to WT (Fig. 5a, c, Supplementary Fig. 9c). As readout of CD4-CTL effector function, we next assessed perforin production and observed significantly increased frequencies of bulk perforin-producing Eomes$^+$T-bet$^+$ and Eomes$^+$T-bet$^-$ cells in th DLN of Aiolos-deficient mice compared to WT (Supplementary Fig. 9d, e). We did not observe a difference in the frequency of Eomes$^-$T-bet$^+$ perforin-expressing cells, further demonstrating a correlation between Aiolos deficiency and Eomes and perforin expression (Supplementary Fig. 9d, e). Consistent with these findings, we also observed an increase in the frequency of antigen-specific NP$^+$Eomes

$^+$perforin$^+$ cells in the absence of Aiolos in the DLN; however, low numbers of antigen-specific cells in the lung of Aiolos-deficient animals precluded reliable assessment of perforin production in this context (Fig. 5d, e, Supplementary Fig. 9f).

As our transcript analyses suggested that Aiolos may function to repress IL-2R subunit expression, we sought to determine whether alterations in IL-2 sensitivity may contribute to augmented CD4-CTL responses in Aiolos-deficient animals. Indeed, Aiolos-deficient CD4$^+$ effector cells exhibited significantly elevated CD25 surface expression relative to WT in both NP-specific and bulk CD4$^+$ effector populations

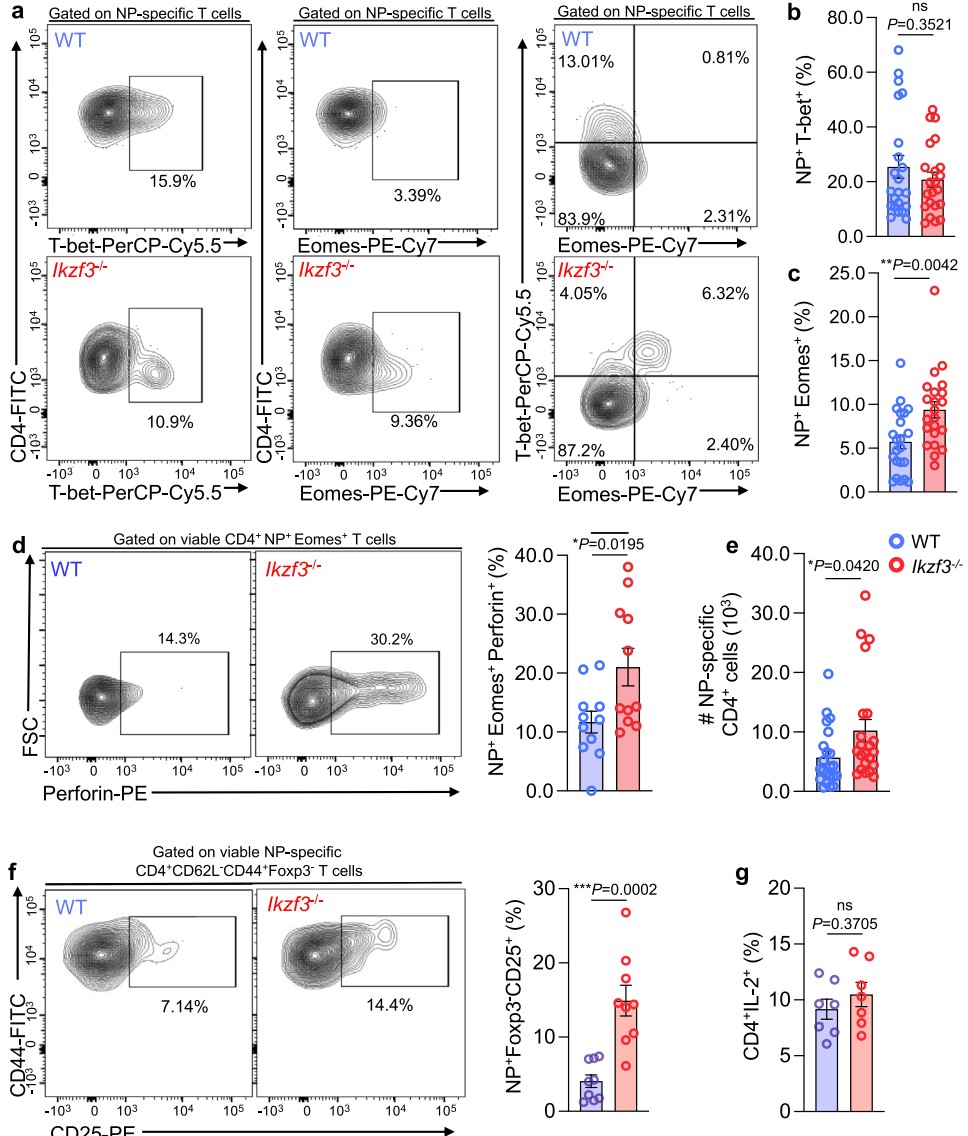

**Fig. 5 | Aiolos deficiency results in acquisition of cytotoxic features by CD4 T cells in vivo.** Naïve WT or Aiolos-deficient (*Ikzf3⁻/⁻*) mice were infected intranasally with 30 PFU influenza (A/PR8/34; "PR8"). After 8 days, draining lymph nodes (DLN) were harvested and viable CD4⁺ effector populations were analyzed via flow cytometry. **a**–**c** Analysis of of T-bet and Eomes expression by influenza nucleoprotein (NP)-specific CD4⁺. Data are compiled from 7 independent experiments ($n = 22 \pm$ s.e.m; **$P < 0.01$; two-sided, unpaired Student's *t* test). **d** Single-cell suspensions from the DLN were incubated in culture medium in the presence of protein transport inhibitors for 3 h. Perforin production by Eomes⁺ populations was analyzed via flow cytometry. Data are compiled from four independent experiments ($n = 11 \pm$ s.e.m; *$P < 0.05$, **$P < 0.01$; two-sided, unpaired Student's *t* test). **e** CD4⁺ NP-specific cells were enumerated. Data are compiled from 7 independent experiments ($n = 22 \pm$ s.e.m; *$P < 0.05$; two-sided, unpaired Student's *t* test). **f** CD25 (IL-2Rα) surface expression was evaluated on viable antigen-specific CD4⁺CD44⁺CD62L⁻Foxp3⁻ (effector) T cells from the lung-draining lymph nodes (DLN) via flow cytometry. Percentage of CD25⁺ cells are shown. Data are compiled from three independent experiments ($n = 9 \pm$ s.e.m; **$P < 0.01$, ****$P < 0.0001$; two-sided, unpaired Student's *t* test). **g** Production of IL-2 by CD4⁺ T cell populations was analyzed in WT vs. Aiolos-deficient cells from the DLN by flow cytometry. Cells were cultured in the presence of PMA and Ionomycin and protein transport inhibitors for 3 h prior to analysis. Data are compiled from two independent experiments ($n = 7 \pm$ s.e.m; two-sided, unpaired Student's *t* test). Source data are provided as a Source Data file.

(Fig. 5f, Supplementary Fig. 9g). We considered the possibility that the increase in CD25 surface expression may be due to an overall increase in IL-2 production in the absence of Aiolos, as Aiolos was previously shown to repress IL-2 in in vitro-generated T_H17 cells[39]. However, we observed no significant difference in IL-2 production between WT and Aiolos-deficient CD4⁺ T cell populations, suggesting that Aiolos may function to repress IL-2 responsiveness, rather than the expression of IL-2 itself (Fig. 5g). Collectively, these in vitro and in vivo findings indicate that Aiolos restrains expression of a cytotoxic-like program, at least in part, via repressive effects on the T-box transcription factor Eomes and IL-2/STAT5 signaling.

## Aiolos-deficient cells exhibit CD4-CTL hallmark features in a CD4⁺ T cell-intrinsic manner

In contrast to the roughly equivalent numbers of WT and Ikzf3⁻/⁻ NP-specific cells in the DLN, we observed a marked reduction of NP-specific cells in lungs of Aiolos-deficient mice relative to WT (Fig. 5e, Supplementary Fig. 9d). While these data are interesting in that they are suggestive of a potential migratory issue, they also precluded reliable analyses of CD4-CTLs in the lungs of Aiolos-deficient mice. Therefore, to both analyze increased numbers of antigen-specific cells in the lung, as well as to evaluate the CD4⁺ T cell-intrinsic role of Aiolos in regulating CD4-CTL responses, we employed an adoptive transfer

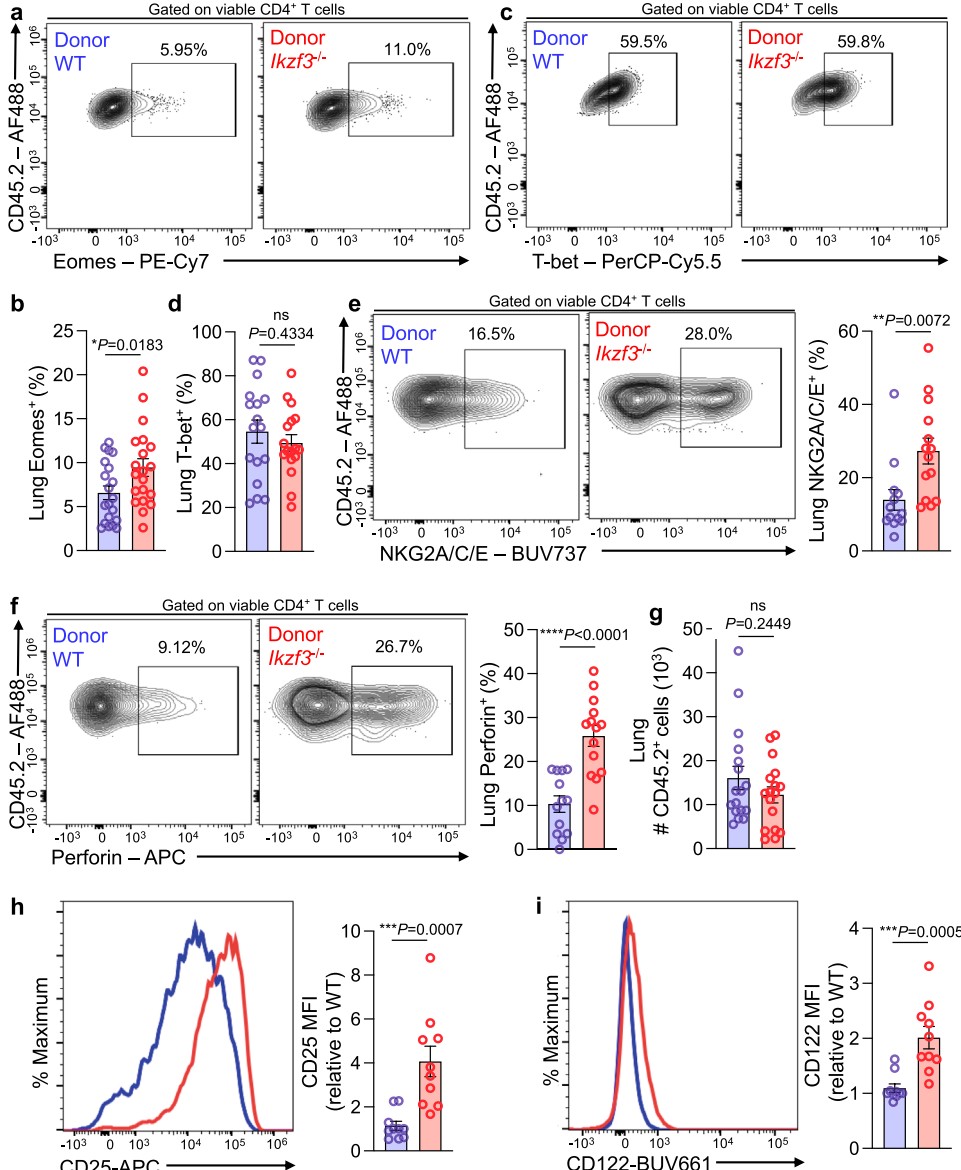

**Fig. 6 | Aiolos-deficient CD4+ T cells exhibit increased cytotoxic hallmarks in an otherwise wildtype setting.** Naïve CD4+ T cells were isolated from OT-II-WT and OT-II-$Ikzf3^{-/-}$ mice. $5 \times 10^5$ cells were adoptively transferred into naïve CD45.1 mice which were then infected with 40 PFU OVA$_{323-339}$-expressing A/PR8/34 ("PR8-OT-II"). After 8 days, lungs were harvested and viable CD45.2+CD4+ populations were analyzed via flow cytometry. **a, b** Analysis of Eomes expression in antigen-specific (CD45.2+CD4+) cells. Data are representative of 6 independent experiments ($n = 20 \pm$ s.e.m; *$P < 0.05$, two-sided, unpaired Student's $t$ test). **c, d** Analysis of T-bet expression in antigen-specific (CD45.2+CD4+) cells. Data are representative of five independent experiments ($n = 17 \pm$ s.e.m; two-sided, unpaired Student's $t$ test). **e–i** Whole-tissue homogenates from the lung were stimulated ex vivo in culture media with OVA$_{323-339}$ peptide for 48 h. Suspensions were then incubated in the presence of protein transport inhibitors for 3 h. **e, f** Flow cytometry analyses of antigen-specific (CD45.2+CD4+) cells for the expression of NKG2A/C/E and perforin. Data are representative of 4 independent experiments ($n = 13 \pm$ s.e.m; **$P < 0.01$, ****$P < 0.0001$, two-sided, unpaired Student's $t$ test). **g** Analysis of the number of antigen-specific (CD45.2+CD4+) cells from whole lung tissue. Data are representative of 5 independent experiments ($n = 17 \pm$ s.e.m; two-sided, unpaired Student's $t$ test). **h, i** Flow cytometry analyses of ex vivo stimulated antigen-specific (CD45.2+CD4+) cells for the expression of CD25 (IL-2Rα) and CD122 (IL-2Rβ). Data are representative of three independent experiments ($n = 10 \pm$ s.e.m; ***$P < 0.001$, two-sided, unpaired Student's $t$ test). Source data are provided as a Source Data file.

strategy[28]. We adoptively transferred naïve CD45.2+ CD4+ T cells from either WT OT-II or $Ikzf3^{-/-}$ OT-II mice into naïve CD45.1+ recipients. Recipient mice were infected with OVA$_{323-339}$-expressing PR8 (PR8-OVA) for 8 days. Analysis of adoptively transferred populations in the lung were consistent with our observations in germline knockout animals, and revealed a significant increase in the frequency of both Eomes+CD4+ T cells and CD25+CD4+ T cells in Aiolos-deficient donor cells as compared to WT (Fig. 6a, b and Supplementary Fig. 10a). As before, T-bet+ populations remained unchanged (Fig. 6c, d). To assess Aiolos-dependent alterations in functional cytotoxic features, we stimulated total lung homogenate ex vivo with OVA$_{323-339}$ peptide.

Consistent with a role for Aiolos in repressing acquisition of the CD4-CTL program in a T cell-intrinsic manner, Aiolos-deficient cells exhibited increased expression of NKG2A/C/E, a surface marker for CD4-CTL populations[4,46–48] (Fig. 6e). Further, Aiolos-deficient donor cells produced significantly more IFN-γ, granzyme B, and perforin relative to their wildtype counterparts (Fig. 6f, Supplementary Fig. 10b–d). In contrast to germline knockout samples, we observed no significant differences in the number of antigen-specific cells between WT and Aiolos-deficient donor populations in the lung (Fig. 6g). To determine whether the effect of Aiolos deficiency on IL-2-responsiveness was similarly CD4+ T cell-intrinsic, we examined surface expression of CD25

(IL-2Rα) and CD122 (IL-2Rβ). Indeed, we observed augmented surface expression of both CD25 and CD122 on Aiolos-deficient cells relative to their WT counterparts (Fig. 6h, i). Collectively, these data support a role for Aiolos in repressing cytotoxic programming in a CD4+ T cell-intrinsic manner, likely by reducing their ability to respond to IL-2/STAT5 signals.

Curiously, unlike our findings in the germline knockout setting, we did not observe defects in T_FH cell differentiation in adoptively transferred Aiolos-deficient cells in the DLN, though regulatory populations were unchanged in this setting (Supplementary Fig. 10e–g). Further, we did not observe altered T-bet or Eomes expression in Aiolos-deficient cells in the DLN, suggesting that alterations to the DLN microenvironment, TCR signaling (polyclonal vs. monoclonal), or antigen availability may contribute to the differing phenotypes observed in this model (Supplementary Fig. 10h, i).

### Aiolos deficiency results in increased chromatin accessibility at regulatory regions for key CD4-CTL genes

To elucidate mechanisms by which Aiolos may function to repress cytotoxic programming, we performed ATAC-seq analyses on WT and Aiolos-deficient cells generated under T_H1-polarizing conditions. As with T_FH cells, PCA analyses revealed distinct genotype clusters (Supplementary Fig. 11a). We observed global changes in chromatin accessibility as well as statistically significant increases in chromatin accessibility at numerous CD4-CTL gene regulatory regions (Fig. 7a, Supplementary Fig. 11b, c). Consistent with augmented transcript expression in the absence of Aiolos, these included regulatory regions for genes encoding cytokine receptors (*Il2ra, Il2rb*), key transcription factors (*Prdm1, Eomes*) and additional CD4-CTL genes, including *Prf1, Eomes, GzmB, Ifng, Il2rb,* and *Klrk1* (which encodes NKG2A) (Fig. 7a–f, Supplementary Fig. 12–14). Motif analysis revealed that, as in T_FH cells, sites of increased accessibility in the absence of Aiolos were enriched for motifs for the CTL transcriptional activators T-bet and Eomes, while sites of decreased accessibility were enriched for Blimp-1 binding motifs (Fig. 7g, h). Curiously, analysis of publicly available STAT5 ChIP seq data revealed that many regions of significantly altered accessibility overlapped with sites of STAT5 enrichment in cytotoxic T cells (Fig. 7b–f, Supplementary Fig. 12–14). Motif analysis corroborated these observations, as sites of increased accessibility in the absence of Aiolos were enriched for STAT5 binding motifs (Fig. 7g, h, Supplementary Fig. 12–14). Together, these data suggest that loss of Aiolos results in significant alterations to accessibility at numerous CD4-CTL gene regulatory regions, and that Aiolos may function to antagonize STAT5 activity at these loci.

### Aiolos deficiency results in augmented STAT5 activity at key CD4-CTL gene loci

We next sought to determine how loss of Aiolos may impact both IL-2 sensitivity and downstream STAT5 activity. Our data suggested that loss of Aiolos resulted in augmented IL-2R transcript expression (Figs. 2a, b, 4g, Supplementary Fig. 15a). To determine if Aiolos overexpression alone was sufficient to repress IL-2R subunit expression, we overexpressed Aiolos in T_H1-polarized cells. Indeed, overexpression of Aiolos resulted in significantly reduced transcript expression for both *Il2ra* and *Il2rb* (Supplementary Fig. 15b). To see whether loss of Aiolos would result in alterations to STAT5 activity, we first examined STAT5 tyrosine phosphorylation, which is indicative of activation/dimerization, in both T_H1- and T_FH-polarized cells. Indeed, we found that Aiolos deficiency resulted in increased STAT5 activation in both cell types, relative to WT (Fig. 8a, Supplementary Fig. 15c). To determine whether Aiolos was directly or indirectly modulating *Il2ra* expression, we used ChIP to assess Aiolos enrichment in T_FH populations, where Aiolos expression is highest (Fig. 1a). Indeed, we found that Aiolos was significantly enriched *Il2ra* promoter when compared to a control region,

indicating that Aiolos may directly repress its expression (Supplementary Fig. 15d).

Finally, we wanted to determine whether Aiolos deficiency resulted in enhanced STAT5 binding to CD4-CTL target genes, particularly at sites where alterations in chromatin accessibility overlapped with STAT5 ChIP-seq peaks (Fig. 7b–d and Supplementary Fig. 12–14). We performed ChIP for STAT5 in WT and Aiolos-deficient cells cultured under T_H1 conditions. Consistent with STAT5-mediated induction, STAT5 enrichment was increased at the *Il2ra* promoter in Aiolos-deficient cells, with no change observed in a control region (Fig. 8b). Similar analyses revealed enhanced STAT5 enrichment at regulatory regions of genes encoding CTL transcription factors (*Eomes, Prdm1;* Fig. 8c) and effector molecules (*Ifng, Gzmb,* and *Prf1;* Fig. 8d) in Aiolos-deficient cells, which each exhibited enhanced chromatin accessibility and expression in this setting (Fig. 4a–g, Fig. 7a–f, Supplementary Fig. 12–14). Finally, analysis of the positive chromatin mark H3K27Ac at these sites revealed significant increases in its enrichment in the absence of Aiolos, suggesting that these loci are more transcriptionally active in Aiolos-deficient settings (Fig. 8e). Collectively, the above findings support a mechanism whereby Aiolos represses IL-2Rα expression, STAT5 activation, and STAT5 association with CD4-CTL gene targets to suppress the CD4-CTL gene program (Supplementary Fig. 16).

## Discussion

CD4-CTLs have emerged as significant mediators of immunity during infection, anti-tumor responses, and autoimmunity. Despite their documented importance, the mechanisms that promote CD4-CTL differentiation have remained somewhat enigmatic[3,4]. In this study, we establish Aiolos as a negative regulator of CD4-CTL programming. Of note, we find that Aiolos deficiency in both T_FH- and T_H1-polarized populations leads to upregulation of a CD4-CTL-like gene program. The origin of CD4-CTL populations has been debated, with some studies suggesting that numerous effector CD4+ T cell subsets have the ability to express cytotoxic features, while others suggest that commitment to a cytotoxic lineage occurs shortly after naïve T cell activation[49,50]. Our findings are in line with work suggesting that CD4-CTLs can arise from T_H1 cells, which some studies have proposed as the most common origin of CD4-CTL populations due to partially shared programming (e.g. IL-2/STAT5, Blimp-1, T-bet)[4,51–53]. Thus, our findings suggest that Aiolos may function as a molecular safeguard to maintain T_H1 (and possibly T_FH) programming from transitioning to a cytotoxic state.

Mechanistically, our findings suggest that enhanced CTL activity in the absence of Aiolos is regulated at least in part by augmented IL-2/STAT5 signaling (Supplementary Fig. 16). We find that loss of Aiolos results in increased IL-2 responsiveness and, consequently, STAT5 association with its target genes. As IL-2/STAT5 signaling positively regulates numerous CD4+ T cell subsets, including both T_H1 and CD4-CTL populations, it is interesting to note that our data implicate Aiolos predominantly in the repression of CTL responses in vivo. Notably, we do not observe augmented IL-2 production by Aiolos-deficient cells, suggesting that Aiolos-dependent regulation of IL-2R subunits occurs independently from IL-2 production itself. These findings are distinct from a previous study implicating Aiolos in the direct repression of IL-2 production in T_H17 populations[39]. This ultimately supported T_H17 differentiation, which is repressed by IL-2 signaling[54]. This difference is notable, as unlike T_H17 cells, T_FH cells maintain the ability to produce IL-2, highlighting an important difference for the role of Aiolos in promoting T_H17 and T_FH differentiation. Further, we find that loss of Aiolos leads to significantly elevated expression of both IL-2Rα (CD25) and IL-2Rβ (CD122) subunits, thus enhancing the ability of differentiating cells to respond to IL-2 signals. CD122 also represents a component of the IL-15R complex, suggesting that the impact of Aiolos may extend to IL-15/STAT5 signaling[55]. This may be especially relevant

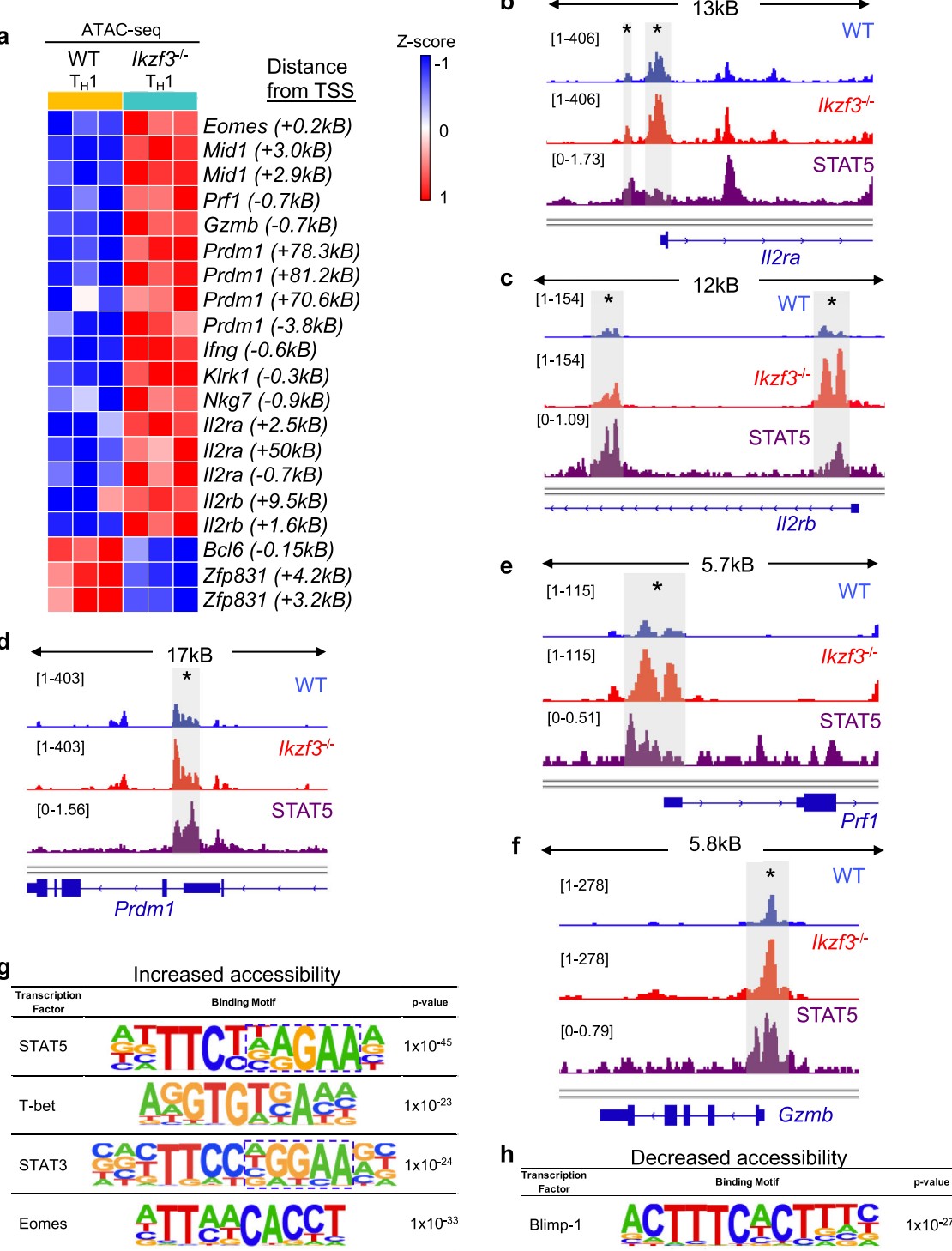

**Fig. 7 | Aiolos deficiency results in increased chromatin accessibility at regulatory regions for key CD4-CTL genes.** Naïve WT and Aiolos-deficient CD4+ T cells were cultured in the presence of $T_H1$-polarizing conditions for 3 days. Assay for Transposase-Accessible Chromatin (ATAC)-seq was performed to assess changes in accessibility at differentially expressed gene loci. **a** Representative heatmap of significantly differentially accessible regions associated with CD4-CTL and $T_{FH}$ gene programs in WT vs Aiolos-deficient cells, presented as row (gene) Z-score with distance from TSS indicated. Regions shown each exhibit statistically significant differences in accessibility. Data are compiled from three independent experiments. **b**–**f** ATAC-seq analyses of $T_H1$ samples overlaid with published STAT5 ChIP-seq (GSM1865310; dark purple) data. WT (light blue, top track) and Aiolos-deficient (red, middle track) samples are displayed as CPM-normalized Integrative Genomics Viewer (IGV) tracks (representative from three independent experiments). Regulatory regions of significant differential accessibility are indicated by gray boxes and asterisks. **g**, **h** HOMER motif analysis of sites of significantly increased (**g**) and decreased (**h**) accessibility in the absence of Aiolos. Data are compiled from 3 independent experiments ($n = 3$, HOMER hypergeometric analysis). Source data are provided as a Source Data file.

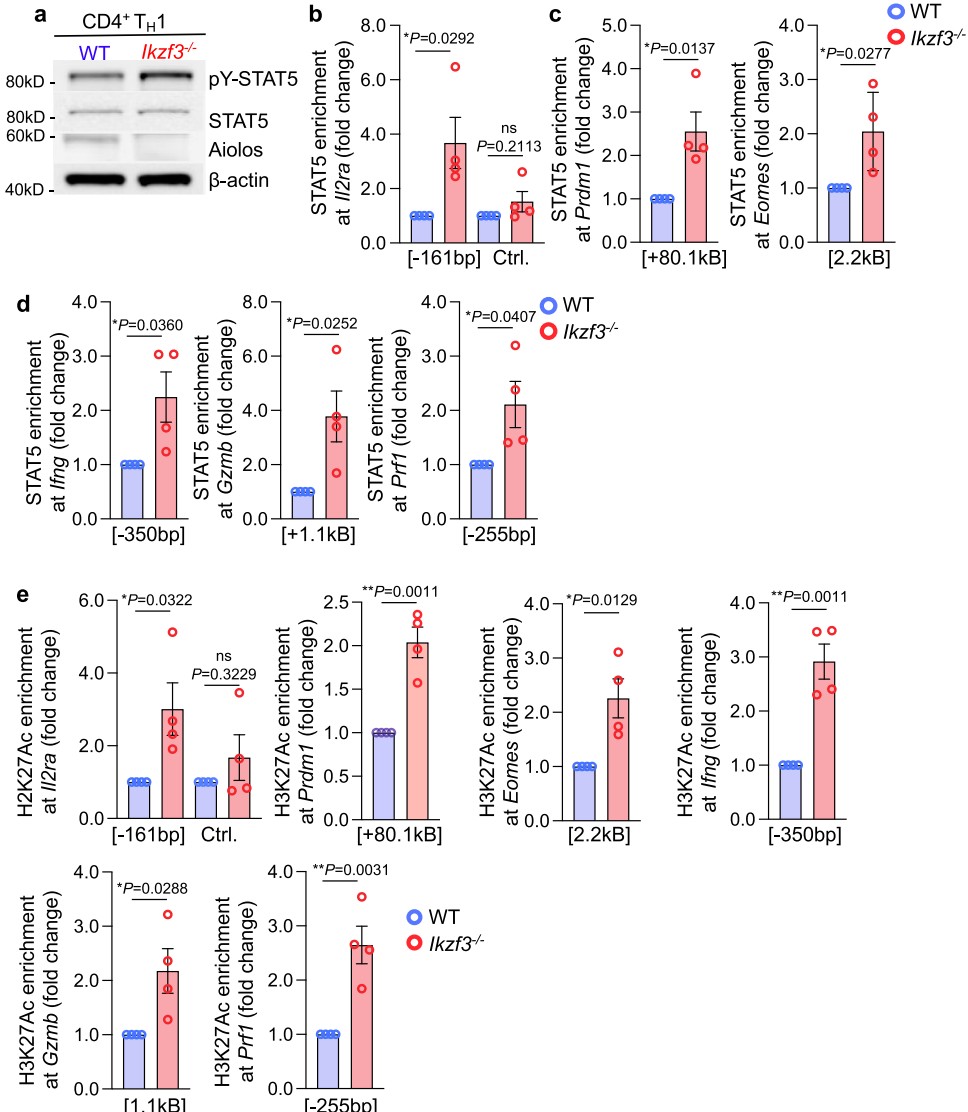

**Fig. 8 | Aiolos deficiency results in increased STAT5 binding and positive chromatin marks at key CD4-CTL transcription factors and effector molecules.** Naïve WT and Aiolos-deficient CD4$^+$ T cells were cultured in the presence of T$_H$1-polarizing conditions for 3 days. **a** Immunoblot analysis of the indicated proteins. β-actin serves as a loading control. A representative image from 4 independent experiments is shown. **b–e** Naïve WT and Aiolos-deficient CD4$^+$ T cells were cultured in the presence of T$_H$1-polarizing conditions for 4 days. ChIP analysis was performed using anti-STAT5, anti-H3K27Ac, and IgG control antibodies for the indicated regions. Data are normalized to total input sample. IgG values were subtracted from percent enrichment and data are displayed relative to the WT sample. Distance from TSS are indicated. Data are compiled from three independent experiments ($n = 4 \pm$ s.e.m; *$P < 0.05$; **$P < 0.01$; two-sided, unpaired Student's $t$ test). Source data are provided as a Source Data file.

in tissues such as the lung, where signals from IL-15 could impact CD4-CTL responses[56]. As both IL-2 and IL-15 signals have been implicated in the repression of T$_{FH}$ programming, it is interesting to speculate that they may also be directly involved in the repression of Aiolos expression itself[13,32,34,35,57–59]. Regardless, our findings are in line with an increasing body of literature identifying roles for IkZF family members in the regulation of cytokine signaling pathways[43].

ATAC-seq analyses revealed that (1) alterations in accessibility in the absence of Aiolos often overlapped with known sites of STAT5 enrichment, and (2) both STAT3 and STAT5 DNA binding motifs are enriched at sites of differential accessibility between WT and Aolos-deficient cells. Strikingly, these motifs appear to contain the core Ikaros zinc finger DNA binding sequence[60]. These findings are significant for two reasons. First, we previously established that Aiolos interacts and cooperates with STAT3 to directly induce the expression of *Bcl6*[38]; thus, overlapping Aiolos/STAT3 motifs at numerous differentially accessible regions suggest that these factors may cooperate to

broadly induce T$_{FH}$ programming. Second, numerous prior studies have demonstrated that competitive mechanisms between STAT3 and STAT5 are important for the specification of CD4$^+$ T cell subset programming, including T$_H$1, T$_H$17, and T$_{FH}$ populations[13,32,33,35,37,54,61,62]. Thus, the overlapping motifs we observe are also suggestive of potential Aiolos or Aiolos/STAT3 antagonism with STAT5. These findings are consistent with several recent studies suggesting that a related family member, Ikaros, which shares the core IkZF DNA binding motif, engages in antagonism with STAT5 in both normal B cell development and leukemic settings[44,45]. Ultimately, further work is needed to establish whether the predominant role of Aiolos is to repress IL-2R subunit expression to alter STAT5 activity, or whether Aiolos–possibly in cooperation with STAT3–functions to antagonize STAT5 binding more broadly.

Our data do support a role for Aiolos in repressing CTL programming via direct induction of Bcl-6 and Zfp831. The latter has recently been implicated in the induction of both Bcl-6 and TCF-1,

which, in addition to promoting $T_{FH}$ differentiation, also are known to repress CD4-CTL programming[21]. It is interesting to note that Aiolos may function as a conserved regulator of these pathways in both $T_H1$ and $T_{FH}$ populations, as our RNA-seq analyses revealed reduced expression of *Bcl6* and *Zfp831* in both cell types. As loss of Aiolos in $T_H1$ cells results in the expression of a CTL-like program, it is interesting to speculate that a certain degree of Aiolos expression may be required in $T_H1$ populations to suppress CTL gene expression. Indeed, our previous work suggests that Bcl-6 is important in $T_H1$ populations to maintain appropriate levels of *Ifng* expression and repress alternative gene programs, through physical interaction with and recruitment by T-bet[63]. As such, it is possible that the absence of Aiolos in this context contributes to augmented cytotoxic features due to the loss of Bcl-6-mediated repression[63].

Finally, our data implicate Aiolos as a direct, positive regulator of $T_{FH}$ transcription factors in vitro, supporting a CD4+ T cell-intrinsic role for Aiolos in regulating the $T_{FH}$ cell differentiation program. Consistent with this, global deficiency of Aiolos resulted in defective $T_{FH}$ cell responses and pathogen-specific antibody production during responses to influenza infection in vivo. These findings are in line with a recent study implicating a dominant-negative Aiolos mutant in the development of numerous T and B cell abnormalities in humans, including disrupted $T_{FH}$ differentiation and germinal center responses[64]. These data, together with our findings, suggest that normal Aiolos expression/function is necessary for effective humoral immunity. However, it is important to acknowledge that the adoptive transfer analyses in our study did not reveal a CD4+ T cell-intrinsic defect in $T_{FH}$ cell differentiation in the absence of Aiolos during influenza infection. As Aiolos has been implicated in the regulation of B cell populations, which engage in supportive bi-directional signaling with cognate $T_{FH}$ cells, we cannot rule out the possibility that defects in B cells affect $T_{FH}$ cell populations in Aiolos-deficient animals. Yet, our findings also suggest that additional (or alternative) mechanisms may be at play. First, we observed substantial alterations in regulatory CD4+ T cell populations in the absence of Aiolos, including both Foxp3+ $T_{REG}$ cells, which are significantly elevated in the absence of Aiolos, and CD4+ Foxp3+Cxcr5+ $T_{FR}$ cells, which are significantly reduced. It is possible that these disrupted regulatory populations may impact $T_{FH}$ differentiation—a defect which may be rescued during adoptive transfer, as healthy populations of both of these cell types are present in WT recipient animals[65,66]. Second, analysis of the lung-draining lymph node during adoptive transfer studies revealed a lack of clear phenotype for both $T_{FH}$ and CD4-CTL populations, yet transferred populations in the lung recapitulated the enhanced CD4-CTL responses we observed in the global Aiolos knockout setting. This suggests that the use of OT-II TCR-transgenic transferred cell populations may contribute to changes in the DLN microenvironment that result in different phenotypes in adoptive transfer, which have been previously observed[67]. All of these are important considerations when interpreting our adoptive transfer findings, and further work will be necessary to assess the T cell-intrinsic role of Aiolos in regulating $T_{FH}$ responses.

It should be noted that Aiolos mutations have been implicated in numerous human diseases, including both autoimmune disease and lymphoid cancers. Therapeutically, Aiolos, along with the IkZF family member Ikaros, is a target of the FDA-approved chemotherapeutic lenalidomide, which is used for the treatment of multiple myeloma[68]. Lenalidomide has also been explored in clinical trials for the treatment of autoimmune disorders including systemic lupus erythematosis, as its immunomodulatory activities are well-established[69,70]. Further, exposure of CAR-T cells to lenalidomide has been shown to potentiate cytotoxic anti-tumor activities of these cells in murine cancer models, though the mechanisms are still being investigated[71]. Thus, our findings provide needed clarity into the potential mechanisms by which Aiolos functions in both healthy and dysregulated immune cell populations. Ultimately, this work provides insight into both $T_{FH}$ and

CD4-CTL biology, and highlights the potential of Aiolos as a targeted therapeutic for the manipulation of CD4+ T cell-dependent humoral and cytotoxic responses.

## Methods
Our research complies with all relevant ethical regulations as required by the Institutional Biosafety Committee (IBC), Institutional Review Board (IRB), and Institutional Animal Care and use Committee (IACUC) of The Ohio State University.

### Mouse strains
CD45.1+ and CD45.2+ C57BL/6 mice were obtained from the Jackson Laboratory. Aiolos-deficient mice were originally obtained from Riken BRC and backcrossed to the C57BL/6 Jackson background for more than 10 generations to generate *Ikzf3−/−*/J mice (herein referred to as "*Ikzf3−/−*"). OT-II mice (with the transgene located on the y-chromosome, originally generated by the Carbone laboratory[72]) were a generous gift of Dr. Haitao Wen. *Ikzf3−/−* mice were crossed to OT-II mice to generate OT-II *Ikzf3−/−* animals for adoptive transfer studies. Germline knockout studies involved the use of both male and female mice. As the OT-II transgene was located on the y-chromosome, adoptive transfer studies utilized only male mice (donors and recipients). For each individual experiment and replicate, mice were age- and sex-matched. All studies performed on mice were done in accordance with the Institutional Animal Care and Use Committee at the Ohio State University in Columbus, OH, which approved all protocols used in this study.

### CD4+ T cell isolation and culture
Naïve CD4+ T cells were isolated from the spleens and lymph nodes of 5–7 week old mice using the BioLegend Mojosort naïve CD4+ T cell isolation kit according to the manufacturer's recommendations. Naïve cell purity was verified by flow cytometry and routinely exceeded 96-98%. For the in vitro polarization of $T_{FH}$-like and $T_H1$ populations, cells ($1.5–2 \times 10^5$ cells/mL) were cultured in complete IMDM ((IMDM [Life Technologies], 10% FBS [26140079, Life Technologies], 1% Penicillin-Streptomycin [Life Technologies], and 50 µM 2-mercapto-ethanol [Sigma-Aldrich]) on plate-bound anti-CD3 (clone 145-2C11; 5 µg/mL) and anti-CD28 (clone 37.51; 2 µg/mL) in the presence of IL-4 neutralizing antibody (clone 11B11, BioLegend, 5 µg/mL) for 18–20 h before the addition of cytokines as follows: for $T_{FH}$ polarization: rmIL-6 (R&D, 100 ng/mL); for $T_H1$ polarization, rmIL-12 (R&D, 5 ng/mL) and rhIL-2 (NIH, 150 U/mL). For $T_H2$ polarization, cells were cultured on plate-bound anti-CD3 (5 µg/mL) and anti-CD28 (2 µg/mL) in the presence of IFN-γ neutralizing antibody (XMG1.2, BioLegend, 10 µg/mL) and rmIL-4 (R&D, 10 ng/mL). For $T_H17$ polarization, cells were cultured on plate-bound anti-CD3 (5 µg/mL) and anti-CD28 (2 µg/mL) in the presence of IFN-γ-neutralizing antibodies (10 µg/mL), IL-4 neutralizing antibodies (5 µg/mL), IL-6 (50 ng/mL), and TGF-β (Gibco, 3 ng/mL). For the generation of $T_H0$ cells, cells were cultured as above in the absence of cytokines and neutralizing antibodies. For some experiments, IL-2 was also neutralized (JES6-1A12, BioLegend, 5 µg/mL). Cells were cultured on stimulation for 72–96 h before analysis. For analysis of cytotoxic effector molecule production, cells were treated with protein transport inhibitors while on stimulation for 3 h prior to analysis by flow cytometry. For overexpression of Aiolos in primary murine T cell populations, the Aiolos (*Ikzf3*) coding sequence was subcloned into the pMSCV-IRES–GFP II (pMIG II, Addgene 52107) vector backbone to generate pMIG-Aiolos. The cloned *Ikzf3* sequence was verified by sequencing and overexpression was validated via both qRT-PCR and immunoblot using an anti-Aiolos antibody. The Platinum-E (Plat-E) Retroviral Packaging Cell Line (Cat# RV-101, Cell Biolabs, Inc.) was used to package pMIG-Aiolos virus per the manufacturer's instructions. Viral supernatant was added to primary murine T cells on α-CD3/α-CD28 stimulation as above (1:1 vol/vol ratio). Cells were transduced in

the presence of 8 µg/mL polybrene (Sigma-Aldrich) using Spinfection for 2 h at 800xg at room temperature. Medium was replaced after 2 h, and transduced cells were collected after 48 h of culture for analysis.

## RNA isolation and qRT-PCR

Total RNA was isolated from the indicated cell populations using the Macherey-Nagel Nucleospin RNA Isolation kit as recommended by the manufacturer. cDNA was generated from mRNA template using the Superscript IV First Strand Synthesis System with provided oligo dT primer (Thermo Fisher). qRT-PCR reactions were performed with the SYBR Select Mastermix for CFX (ThermoFisher) using 10–20 ng cDNA per reaction (primers provided in Supplementary Table 1). All qRT-PCR was performed on the CFX Connect (BioRad). Data were normalized to *Rps18* and presented either relative to *Rps18* or relative to the control sample, as indicated.

## Influenza infection and tissue preparation

Influenza virus strain A/PR8/34; "PR8" was propagated in 10-day-old embryonated chicken eggs and titered on MDCK cells (BEI Resources, NIAID, NIH: Kidney (Canine), Working Cell Bank, cat# NR-2628). Naïve mice were infected intranasally with 30 PFU PR8. After 8 days, draining lymph nodes were harvested. For serum antibody analyses, whole blood was collected by cardiac puncture for serum isolation. For DLN, single-cell suspensions were generated in tissue preparation media (IMDM + 4% FBS) by passing tissue through a nylon mesh strainer, followed by erythrocyte lysis via 3-minute incubation in 0.84% $NH_4Cl$. For lung single-cell suspension preparation, whole lungs were dissociated in Collagenase IV containing preparation media via a GentleMACS Dissociator (Miltenyi Biotech) according to the manufacturer's instructions. Dissociated tissue was then passed through a nylon mesh strainer. Cells were layered with Percoll in RPMI and centrifuged at 500xg for 20 min at room temperature with the brake off. The mononuclear layers were harvested and erythrocyte lysis was performed via 3-minute incubation in 0.84% $NH_4Cl$. For analysis of IL-2 production, homogenized samples were cultured in the presence of PMA and Ionomycin and protein transport inhibitors for 3 h prior to staining. For analysis of granzyme B and perforin in polyclonal populations, whole-tissue homogenized samples were treated with protein transport inhibitors for 3 h prior to analysis by flow cytometry. For adoptive transfer studies, naïve CD45.2+ OT-II CD4+ T cells were purified from Ikzf3+/+ wildtype OT-II or Ikzf3−/− OT-II mice using negative selection as described above. Cells were washed 1-2X and resuspended in sterile 1X PBS for retro-orbital transfer ($5 \times 10^5$ cells/animal) into wildtype CD45.1+ recipient mice. After 24 h, mice were infected intranasally with OVA$_{323-339}$-expressing PR8 ("PR8-OVA"). For ex vivo peptide stimulation of OT-II cells, whole-tissue homogenates were processed as above and stimulated with OVA$_{323-339}$ peptide (AnaSpec) at 5 µg/mL for 48 h. At 45 h post-stimulation, homogenized samples were treated with protein transport inhibitors for 3 h prior to analysis by flow cytometry. Cells were washed 1X in ice cold FACS buffer (PBS + 4% FBS) before staining.

## Human tissue samples

Human tissues were collected and utilized in accordance with protocols approved by The Ohio State University Institutional Review Board. Donor consent was acquired when deemed appropriate according to the approved Ohio State University Institutional Review Board protocol. Human pediatric tonsils were obtained following overnight shipment after surgery via the CHTN Western Division at Vanderbilt University (Nashville, TN). Lymphocytes were enriched from fresh tonsil tissue specimens using previously reported protocols[73]. Briefly, single-cell suspensions were generated by dissociation via a GentleMACS Dissociator (Miltenyi Biotech) according to the manufacturer's instructions. Cells were diluted in PBS (Thermo Fisher Scientific), layered over Ficoll-Paque PLUS (GE Healthcare), and centrifuged at

2000 rpm for 20 min at room temperature with the brake off and the mononuclear layers were harvested.

## Flow cytometry

For analysis of antigen-specific CD4+ T cell populations in murine samples, cells were stained in FACS buffer with IA^b NP$_{311-325}$ MHC class II tetramer (1:100, NIH Tetramer Core Facility) at room temperature for 1 h. For extracellular staining, samples were pre-incubated for ≥5 min at 4 °C with Fc Block (clone 93; BioLegend), then stained in the presence of Fc block for 30 min at 4 °C using the following antibodies. For all flow cytometry antibodies, example catalog numbers provided for each antibody, but catalog numbers will vary based on fluorochrome and volume of product: CD4 (1:300; clone GK1.5; R&D Systems; catalog # for anti-CD4:AF488: FAB554G); CD44 (1:300; clone IM7; BD Biosciences; catalog # for anti-CD44:V450: 560451); CD62L (1:300; clone MEL-14; ThermoFisher; catalog # for anti-CD62L-APC-eFluor780: 47-0621-82); PD-1 (1:50; clone 29 F.1A12; BioLegend; catalog # for anti-PE-Cy7: 135216); Cxcr5 (1:50; clone SPRCL5; ThermoFisher; catalog # for anti-Cxcr5:PE:12-7185-82); CD25 (1:25; clone PC61.5; ThermoFisher; catalog # for anti-CD25:PE: 12-0251-82), CD45.2 (1:100; clone 104; Biolegend; catalog # for anti-CD45.2:AF488: 109816), CD122 (1:50; clone 5H4; Biolegend; catalog # for anti-CD122 BUV711: B741537), NKG2A/C/E (1:100; clone 20D5; Biolegend, catalog # for anti-NKG2A/C/E:BUV737: B741808). At the same time, cells were also stained with Ghost V510 or Red 780 viability dye (1:400-1:750; Tonbo Biosciences). Cells were then washed 2X with FACS buffer prior to intracellular staining. For intracellular staining, cells were fixed and permeabilized using the eBioscience Foxp3 transcription factor staining kit (ThermoFisher) for 30 min at room temperature, or overnight at 4 °C. Following fixation, samples were stained with the following antibodies in 1X eBiosciences permeabilization buffer for 30 min at room temperature: T-bet (1:100; clone 4B10; BioLegend; catalog # for anti-T-bet:PerCP-Cy5.5: 644806); Foxp3 (1:300; clone FJK-16s; ThermoFisher; catalog # for anti-Foxp3:PerCP-Cy5.5: 45-5773-82); Bcl-6 (1:20; clone K112-9; BD Biosciences; catalog # for anti-Bcl-6:AF488: 561524); Aiolos (1:50; clone S48-791; BD Biosciences; catalog # for anti-Aiolos:AF488: 565266); IL-2 (1:50; clone JES6-5H4; BioLegend; catalog # for anti-IL-2:APC: 503810); Eomes (1:100; clone DAN11MAG; ThermoFisher; catalog # for anti-Eomes:PE-Cy7: 25-4875-82); IFN-γ (1:400; XMG1.2; BioLegend; catalog # for anti-IFNg:BV650: 505831); Granzyme B (1:300; clone GB11; ThermoFisher; catalog # for anti-Granzyme B:PE: GRB04); and Perforin (1:100; S16009A; BioLegend; catalog # for anti-Perforin:APC: 154304). For some experiments, samples were also stained with a panel of excluded (dump gate) antibodies: CD45R/B220 (1:300; clone RA3-6B2; Biolegend, catalog # for anti-D45R/B220:BV510: 103247), CD11b (1:300; clone M1/70; Biolegend; catalog # for anti-CD11b:BV510: 101245), F4/80 (1:300, clone BM8, Biolegend, catalog # for anti-F4/80:BV510: 123135), and CD11c (1:300, clone N418, Biolegend, catalog # for anti-CD11c:BV510: 117337). Cells were washed with 1X eBiosciences permeabilization buffer and resuspended in FACS buffer for analysis. Samples were run on a BD FACS Canto II or BD FACS Symphony and analyzed using FlowJo software (version 10.8.1).

For human tissue analysis, lymphocyte populations were stained using antibodies directed against surface or intracellular proteins according to the manufacturers' recommendations. The LIVE/DEAD Fixable Aqua Dead Cell Stain Kit (Thermo Fisher Scientific) was used to exclude nonviable cells in the analysis. Intracellular staining was performed using the Transcription Factor/FOXP3 Fixation/Permeabilization Solution Kit (Thermo Fisher Scientific) according to the manufacturer's instructions. For human sample staining, the following antibodies were used: CD44-APC (1:20; DB105, Miltenyi Biotec, catalog # 130-110-294), CD4-AF700 (1:100; RPA-T4, Thermo Fisher Scientific, catalog # 56-00049-42), CD45RA-APC-Cy7 (1:200; REA1047, Miletnyi Biotec, catalog # 130-117-747), AIOLOS-PE (1:100; 16D9C97, BioLegend, catalog # 371104), CXCR5-PE-Vio770 (1:200; MU5UBEE, ThermoFisher

Scientific, catalog # 25-9185-42), FOXP3-PerCP-Vio770 (1:100; PCH101, ThermoFisher Scientific, catalog # 45-4776-42), CD3-PE-Vio615 (1:100; UCHT1, BioLegend, catalog # 300450), CD8-FITC (1:100; REA734, Miltenyi Biotec, catalog # 130-110-677), PD-1-BV421 (1:200; MIH4, BD Biosciences, catalog # 564323), CD62L-BV605 (1:100; DREG-56, BD Biosciences, catalog # 562720), CD45RO-BV650 (1:200; UCHTL1, Biolegend, catalog # 304232). Samples were run on a FACSAria II (BD Biosciences) and analyzed using FlowJo (BD Biosciences, version 10.8.1).

### Serum antibody ELISA

To detect anti-PR8 antibodies, high-binding microplates (Corning) were coated with UV-inactivated PR8 in PBS at 4 °C overnight. To quantify the absolute antibody concentration, a standard curve for mouse isotype was established for each plate by coating with goat anti-mouse Ig (1 μg/mL in PBS; Southern Biotech) overnight at 4 °C. Plates were blocked with 1% bovine serum albumin (Fisher Scientifics) in PBSTE (Phosphate buffered saline with 0.05% Tween-20 and 1 mM EDTA) for 1 h at 37 °C. After washing with PBSTE, samples and standard Ig (Southern Biotech) were added to respective wells and incubate for 1 h at room temperature (RT). For total IgG standard, a mixture of purified mouse IgG1, IgG2b, IgG2c, and IgG3 were combined to emulate WT antibody titers[74]. Biotinylated goat-anti-mouse isotype-specific (IgM, IgG) antibodies (0.1 μg/mL; Southern Biotech) were added and incubated at RT for 1 h. Horseradish-peroxidase-conjugated streptavidin (ThermoFisher) was added and incubated at RT for 30 min. The plates were washed with PBSTE between steps for the procedures above. TMB High Sensitivity Substrate (BioLegend) was used for detection; reactions were stopped by adding 1 M sulfuric acid. Absorbances were measured at OD450 with OD540 as background using M2e SpectraMax 500 (Molecular Devices) and antibody concentrations were calculated using the generated standard curve.

### RNA-seq analysis

Naïve CD4[+] T cells were cultured under $T_H1$ or $T_{FH}$-polarizing conditions for 3 days. Total RNA was isolated using the Macherey-Nagel Nucleospin kit according to the manufacturer's instructions. Samples were provided to Azenta Life Sciences for polyA selection, library preparation, sequencing, and DESeq2 analysis (3 biological replicates per cell type and genotype, from 3 independent experiments). Genes with a p-value <0.05 were considered significant, and those with fold changes of ≥1.5 were defined as differentially expressed genes (DEGs) for each comparison in the present study. Genes pre-ranked by multiplying the sign of the fold-change by -log10 (p-value) were analyzed using the Broad Institute Gene Set Enrichment Analysis (GSEA) software for comparison against 'hallmark', 'gene ontology', and 'immunological signature' gene sets. Heatmap generation and clustering (by Euclidean distance) were performed using normalized log2 counts from DEseq2 analysis and the Morpheus software (https://software.broadinstitute.org/morpheus). Volcano plots were generated using -log10(p-value) and log2 fold change values from DEseq2 analysis and VolcaNoseR software (https://huygens.science.uva.nl/VolcaNoseR/)[75].

### ATAC-seq analysis

ATAC-seq analysis was performed as described[76]. Briefly, $5 \times 10^4$ cells at >95% viability were processed using the Illumina Nextera DNA Library Preparation Kit, according to the manufacturer's instructions. Resultant sequences were trimmed and aligned to mm10 using Bowtie2. After trimming, all subsequent analyses were performed utilizing the indicated tools in Galaxy (usegalaxy.org). Samples were filtered by read quality (≥30), as well as to remove duplicates and mitochondrial reads. Statistically significant peaks were identified using MACS2 callpeak. DiffBind was used to identify regions of significant differential accessibility between WT and Aiolos-deficient samples for each cell type. Regions with adjusted *P* values <0.05 were considered

statistically significant. CPM-normalized tracks were visualized using Integrative Genomics Viewer (IGV) version 2.12.2. PCA plots were generalized using normalized counts from DiffBind and Clustvis software (https://bio.tools/clustvis). Motif analyses were performed using regions of significantly altered accessibility Aiolos-deficient samples relative to WT using HOMER hypergeometric analysis (http://homer.ucsd.edu/homer/motif/). Heatmap generation and clustering (by Euclidean distance) were performed using normalized counts from DiffBind analyses and the Morpheus software (https://software.broadinstitute.org/morpheus). Volcano plots were generated using -log10(adjusted p-value) and log2 fold change values from DiffBind analyses and VolcaNoseR software (https://huygens.science.uva.nl/VolcaNoseR/)[75].

### Chromatin Immunoprecipitation (ChIP)

ChIP assays were performed as described previously[77]. Resulting chromatin fragments were immunoprecipitated with antibodies against Aiolos (Cell Signaling, clone D1C1E, 2 μg/IP), STAT5 (R&D AF2168, 5 μg/IP), H3K27Ac (abcam ab4729, 1–2 μg/IP) or IgG control (Abcam ab6709; 2–5 μg/IP, matched to experimental antibody). Enrichment of the indicated proteins was analyzed via qPCR (primers provided in Supplementary Table 1). Samples were normalized to total DNA controls and percent enrichment from isotype control antibodies was subtracted from each IP to account for non-specific background.

### Immunoblot analysis

Immunoblots were performed as described previously[38,77]. Briefly, cells were harvested and pellets were lysed and prepared via boiling for 15 min in 1X loading dye (50 mM Tris [pH 6.8], 100 mM DTT, 2% SDS, 0.1% bromophenol blue, 10% glycerol). Lysates were separated by SDS-PAGE on 10% Bis-Tris Plus Bolt gels (ThermoFisher) and transferred onto 0.45 μm nitrocellulose membrane. Membranes were blocked with 2% nonfat dry milk in 1X TBST (10 mM Tris [pH 8], 150 mM NaCl, 0.05% Tween-20), and detection of indicated proteins was carried out using the following antibodies: Aiolos (39293, Active Motif, 1:20,000), Bcl-6 (clone K112, BD Biosciences, 1:500), pSTAT5$_{(Y694/9)}$ (clone 47 BD Biosciences, 1:5000), STAT5 (clone (D206Y, Cell Signaling, 1:5000), β-Actin (GenScript, 1:15,000), Eomes (Abcam, 1:5,000) goat anti-mouse:HRP (Jackson Immunoresearch, 1:5,000-1:30,000), mouse anti-rabbit:HRP (Santa Cruz, 1:5000-1:20,000).

### Statistics and Reproducibility

All statistical analyses were performed using the GraphPad Prism software (version 9.3.1). For single comparisons, two-tailed Student's *t* tests (paired or unpaired, as noted) were performed. For multiple comparisons, one-way ANOVA with Tukey's multiple comparison tests were performed. Error bars indicate the standard error of the mean. *P* values <0.05 were considered statistically significant. All data are representative of at least two independent experiments; all data analyzed statistically are compiled from at least three independent experiments. No statistical method was used to predetermine the sample size. The experiments were not randomized (detailed in the Reporting Summary). The Investigators were not blinded to allocation during experiments and outcome assessment.

### Software summary

Data were collected utilizing the following open-source or commercially available software programs: BD FACSDiva (version 8.0.2), BioRad Image Lab (version 6.0.1, build 34), BioRad CFX Manager (version 3.1). ATAC-seq data was generated by Illumina Novaseq SP. RNA-seq was performed by Azenta Life Sciences (formerly Genewiz). Analyses and/or manuscript preparation were conducted using the Microsoft Office Suite, version 16.43 (including Microsoft Word, Microsoft Excel, and Microsoft Powerpoint), BD FlowJo (version 10.8.1), and open-source software, including tools available on Galaxy (http://usegalaxy.org):

Bowtie2 (2.4.4), MACS2, Integrative Genomics Viewer (version 2.9.4), Morpheus (Broad Institute), VolcanoseR (https://huygens.science.uva.nl/VolcaNoseR/)[75], and ClustVis (https://bio.tools/clustvis). All statistical analyses were performed using the GraphPad Prism software (version 9.3.1). Data preparation for this manuscript did not require the use of custom code or software. Supplementary Fig. 16 was generated using BioRender (https://biorender.com/).

### Reporting summary

Further information on research design is available in the Nature Portfolio Reporting Summary linked to this article.

## Data availability

Data and material availability: RNA- and ATAC-seq datasets have been deposited in the GEO repository under accession number GSE203066. Publicly available data under GSE58597 and GSM1865310 were also analyzed for use in this study. All other datasets and materials from this study will be made available upon reasonable request. Requests should be sent to the corresponding author. Source data are provided with this paper.

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

## Acknowledgements

The authors would like to thank all members of the Oestreich Lab, as well as colleagues in the Department of Microbial Infection and Immunity for constructive criticism. The authors would also like to thank the Cooperative Human Tissue Network of the Nationwide Children's Hospital (Columbus, OH) for providing human pediatric tonsil samples. The authors would like to thank members of the Lio laboratory (Heng-Yi Chen and Fang-Yun Lay) for their technical advice, and Dr. Haitao Wen for providing OT-II animals. Finally, the authors would like to thank members of the Yount laboratory (Ashley Zani, Adam Kenney, and Lizhi Zhang) for assistance with influenza virus preparation. This work was supported by grants from The National Institutes of Health AI134972 and AI127800 to K.J.O., AI156411 to P.L.C., CA199447 and CA208353 to A.G.F., K22CA241290 to C.J.L., F32AI161857 to M.D.P., and R01AI113021 to J.M.B. K.A.R. is supported by funding through The Ohio State University College of Medicine Advancing Research in Infection and Immunity Fellowship Program. J.A.T. is supported by funding through

the Susan Huntington Dean's Distinguished University fellowship. J.A.T. and S.P. are supported by an Interdisciplinary Program in Microbe-Host Biology training grant; 1T32AI165391 awarded by NIH/NIAID. Finally, K.J.O. was also supported by funds from The Ohio State University College of Medicine and The Ohio State University Comprehensive Cancer Center.

## Author contributions

D.M.J. and K.A.R. assisted with the design of the study, performed experiments, analyzed data, and wrote the manuscript. S.P., J.A.T., E.D.S.H., A.V., C.D.E., M.R.L., R.T.W., and A.G.F. performed experiments and analyzed data. P.L.C., M.D.P., and J.M.B. assisted with analysis of -omics data. A.S., O.A., E.A.H, J.S.Y., G.X., H.E.G., and C.J.L. assisted with the adoptive transfer and/or influenza infection experiments and analyzed data. K.J.O. supervised the research, designed the study, analyzed data, and wrote the manuscript.

## Competing interests

The authors declare no competing interests.
