## [Peer Review File · Nature Communications]

Aiolos represses CD4+ T cell cytotoxic programming via reciprocal regulation of TFH transcription factors and IL-2 sensitivityReviewers' comments:

Reviewer #1 (Remarks to the Author):

Summary

Read, Jones and colleagues analyze Aiolos as a regulator of Tfh cell and CD4-CTL responses upon murine flu infection and during in vitro polarization. They find that Aiolos is necessary for Tfh cell development and antibody production in vivo, and in its absence, CD4 T cells assume a CTL fate. They further demonstrate that Aiolos positively regulates a Tfh-cell gene program, including that of Zfp831, TCF-1 and Bcl6, while repressing the expression of IL2Ra and IL-2/STAT5 signaling necessary for expression of the CTL gene program. They conclude that Aiolos serves as a central mediator of Tfh vs CTL development.

Significance

Previous work by the authors demonstrated that Aiolos (Ikzf3) can regulate the expression of Bcl6 in Tfh-like cells, suggesting it promotes their differentiation (Read, J Immunol 2017). The current paper extends this observation, demonstrating that Aiolos positively regulates Tfh cell formation in vivo and in vitro, while reciprocally regulating that of CD4 CTLs, the development of which has not been dissected as done here. These findings are strengths of the paper, alongside dissection of the gene regulatory pathways necessary for the observed phenotypic changes. That said, greater dissection of the phenotypic effects of T-cell intrinsic Aiolos regulation would strengthen the authors' conclusions. While Aiolos target gene expression is investigated, its direct regulatory relationships also could be explored in more depth.

Major points

1. Aiolos deficiency results in disrupted Tfh cell differentiation and antibody production as assessed in Ikzf3 germline deficient mice. While the in vitro experiments suggest this effect is CD4 T cell intrinsic, this is not the case for the in vivo effect. Rather, it could be confirmed using adoptive transfers, bone marrow chimeras, and/or CRISPR gene targeting with adoptive transfer. Such experiments would also better determine Aiolos's functional role in Tfh cells. While the reduced antibody response in germline deficiency is assumed to be secondary to reduced numbers of Tfh cells, their function (trafficking, cytokine production, CD40L expression and so forth) is not tested. Numbers of cells in addition to percentages should be shown throughout. Moreover, while flow data suggest Aiolos is necessary for GC-Tfh cell development, this is an indirect assessment, which would be better determined by GC staining, ideally including for Aiolos. While deficiency in the latter leads to reduction in Ab production 8 days following flu challenge, this time point largely analyzes early PC formation, not GC output, with the latter not determined.

Such experiments would not only rule out a non-T cell intrinsic effect but would eliminate the possibility the phenotype observed is secondary to manipulation of Tfr cells, reduced in the absence of Aiolos. To this point, the authors' reasoning that the observed 'expansion' of regulatory cells rules out a Tfh cell extrinsic effect directly contradicts their observation that Tfr cells are reduced in the germline mutants, an effect which of course can lead to reduced Tfh cell numbers and output. The only way to resolve this point is to do an experiment separating the effects of Aiolos in Tfh vs. Tfr cells.

2. To verify Aiolos targets, the authors used ATAC-seq combined with ChIP-qPCR analysis of WT and mutant cells, finding enrichment in the indicated regions of Tfh-related genes or Th1-associated genes. While compelling, it is unclear which genes are direct Aiolos targets. Also, what is the binding motif for Ikzf3? It is not clear from the data if the binding motif is found in the binding site of different Tfh associated genes.

3. Tcf1 acts as a master transcriptional regulator of T cells, with expression in naïve T cells and maintenance in Tfh cells. Is Tcf1 reduced in naïve cells or only specifically in Tfh cells in Aiolos-deficient mice? Is downregulation of Tcf1 only observed under TFH-like polarizing culture conditions? It would be good to know whether this is specific for Tfh cells. Is the phenotype of Aiolos knockout mice like that of Tcf1 knockout mice, and can overexpression of Tcf1 or other targets rescue the Aiolos-deficient phenotype? Is regulation of Tcf1 by Aiolos universal or specific to Tfh cells?

Minor points

1. Il2ra is increased in Ikzf3^{-/-} cells; however, the effect is relatively small (~1.5-fold), suggesting IL2ra may not be a primary Aiolos target.
2. The paper dissects the “Zfp831/Tcf-1/Bcl6 axis”, yet this is not shown in the graphic model (Extended Data Figure 7).
3. The authors show the Tfh associated gene expression in in vitro cells cultured under Tfh-like polarizing conditions. How about the expression of Tcf1, Bcl6, and Zfp831 in primary Tfh cells from wild-type and Aiolos knockout mice?
4. The Stat5 ChIP-seq peak is missing in the IGV figure (Extended Data Figure 6).

Reviewer #2 (Remarks to the Author):

In this manuscript Read KA and colleagues investigated the role of the transcription factor Aiolos in Tfh cells. In a previous study from the same group (Read KA J Immunol 2017), the authors found that Aiolos expression was increased in antigen-specific Tfh cells after influenza infection, compared to antigen-specific effector T cells. This new study now aims at extending on this finding, by showing that in the absence of Aiolos, Tfh cell differentiation in response to influenza antigens was impaired, leading to reduced antibody response. On the other hand, T cells lacking Aiolos that were cultured under Tfh conditions acquired some features of a cytotoxic-like program, including perforin expression. Aiolos deficiency also resulted in increased CD25 expression.

While some observations warrant further investigation (for example, the role of Aiolos in CD4-CTLs, or in T cell responses to viral infections), other findings appear to be more incremental and preliminary, and in the absence of more thorough and global analyses I remain uncertain whether some of the conclusions are indeed adequately supported by experimental evidence.

- 1) Currently this study is not adequately put into the context of previous relevant work on Aiolos published by other labs. Most notably:
 - Quintana et al (Nature Immunology 2012) showed that Aiolos controlled the differentiation and function of Th17 cells, and that Aiolos was most highly expressed in this subset. In Figure 2a, how does expression of Aiolos in Tfh cells compare to Th17 cells? This information would provide more general insights on its role in T cell differentiation.

- Kuehn et al (J Exp Med 2021) showed that a mutation in AIOLOS identified in human patients was associated to T and B cell abnormalities (including impaired Tfh cell differentiation), recapitulated in a mouse model. In Figure 1, can the authors rule out that reduced antibody production is not due to an intrinsic B cell defect?
- Wang JH (Immunity 1998) showed increased proliferation of Aiolos-deficient T cells in response to TCR activation. Can the authors rule out that some of the differences they observe in the current study are not due differences in T cell activation?

2) Figure 3: since the authors performed ATAC-seq on Tfh cells, they should assess if a more comprehensive analysis (rather than just looking at a few pre-selected genes) provides more information about the regulatory roles of Aiolos in Tfh cells. In general, I don't think that based on a few selected snapshots the authors can draw the conclusion that "chromatin accessibility was negatively impacted by the absence of Aiolos". There might be many more regions where chromatin accessibility is positively impacted. And the slight reduction shown could simply reflect experimental variability. Also, how do these differences compare to differences in accessibility at established Aiolos targets, like Il2?

3) Still in Figure 3, a better control of specificity for the ChIP-PCR assay is represented by the *Ikzf3*^{-/-} Tfh cells, which should be used in comparison with the other experimental groups.

4) The authors could establish if Aiolos indeed directly regulates the expression of some of the identified factors (*Bcl6*, *Il2ra*, etc) by using gain-of-function experiments and luciferase reporter assays.

5) The authors state (page 10) that Aiolos directly regulates CD25 expression. However, the authors only showed that the *Il2ra* transcript is expressed comparatively at higher levels (~2 times higher) in *Ikzf3*^{-/-} Tfh cells compared to wild-type. How does this translate into the dynamics of CD25 expression at different time points of T cell activation and with different strengths of stimulation? A small upregulation of CD25 expression may end up being modest and primarily linked to increased T cell activation, rather than differentiation. B cells lacking Aiolos for example exhibited an activated phenotype (Wang JH Immunity 1998).

6) Along the same line, while I appreciate that the authors show increased CD25 expression in vivo (Figure 7), this could also be due to lower thresholds of T cell activation in absence of Aiolos. What about the surface expression of other activation markers (both in vivo and in vitro) that are not regulated by Aiolos? Including also *Icos*, which is central to Tfh biology.

7) Please also indicate where all the relevant transcripts (including *Il2ra*, *Il2rb*, *Icos*) are located in the volcano plots in Figure 2 and 5.

8) A more in-depth comparison of the RNA-seq data in Tfh and Th1 cells could provide more information about the role of Aiolos in these cells. What is the extent of overlap of the up- and down-regulated genes in Tfh and Th1 conditions in the absence of Aiolos? Some Venn diagrams with indicated common and unique genes would already be informative. If mostly the same genes are dysregulated in these two cell subsets upon Aiolos deletion, then the main phenotype may be linked to general T cell activation rather than differentiation.

9) Figure 6F: was *Cx3cr1* expression increased also in any of the RNA-seq datasets? Excel files containing at the very least the significant differentially expressed genes and differentially accessible regions should be made available to be able to assess the

consistency and quality of the data. The number of biological replicates used in RNA-seq for Tfh cells is not indicated neither in the methods nor in the figure legends. Please show PCAs of all RNA-seq and ATAC-seq experiments. Legend of Figure 3 mentions two replicates for ATAC-seq, which is insufficient to achieve robust results.

10) Based on the CD25 data presented in this manuscript, and in the absence of other information (that is, expression of other activation markers not regulated by Aiolos, functional effects of IL-2 on cell proliferation and so on), the conclusion that “Aiolos represses IL-2 responsiveness” (page 12) is both premature and preliminary, since no direct evidences are presented.

11) I am uncertain what conclusions can be drawn from Figure 8: is the increased STAT5 phosphorylation actually due to IL-2 stimulation? Is it significant? Does it occur also in Aiolos-deficient Tfh cells? Does it affect T cell responses to IL-2?

12) If Aiolos indeed suppresses CD25 expression, then its retroviral transduction in wt and Aiolos-deficient cells should reduce (or normalize, respectively) CD25 expression and any downstream functional effects.

13) The observation that T cells lacking Aiolos express granzyme and perforin is interesting: do they also acquire killing capacity in vitro?

14) It is unclear from the figure legend if the human data shown in Extended Figure 1 show 3 independent donors. I also believe that the graph should show SD and not SEM and paired, rather than unpaired t-test. The authors may want to double-check their statistics throughout the manuscript.

Reviewer #3 (Remarks to the Author):

In the manuscript by Read et al, the authors explore the role of Aiolos in Tfh/CD4 CTL differentiation. In line with their previous publication (Read et al, J. Immunol 2017) the authors find that Aiolos supports Bcl6 and subsequent Tfh differentiation, in this study by using Aiolos (Ikzf3) deficient mice. The loss of Tfh cells leads to a subsequent increase in CD4 T cells that have cytotoxic capabilities, including perforin and Granzyme B, as well as IFN γ . This is attributed to alterations in the chromatin accessibility and binding of Aiolos to Tfh genes, and in contrast, a repressive effect on CD25 leading to increased STAT5 signaling and downstream genes associated with CTL function. While the data are straightforward and good quality, the findings are not surprising in light of prior work from this group and others (Quintana et al, Nat. Immunol. 2012) that Aiolos plays a key role in T cell differentiation by promoting Bcl6, Tfh and inhibiting the IL-2 pathway. There are also several missed opportunities that would bolster the impact of these findings, as well as key deficiencies in experimental design that limit the impact.

1. Most of the data presented are in vitro Th1 and Tfh-like cells. In vivo data with influenza have limited attempts to address functional impact beyond one experiment for antibody formation. In LN, GCs should be assessed if Tfh are decreased. Also missed opportunity to look at CD4-CTLs in the lung tissue. Brown et al (J. Virol. 2012) demonstrated a protective role for CD4-CTLs in influenza infection. Does loss of Aiolos lead to increased viral clearance in the lungs, or perhaps increased immunopathology? What are the functional

consequences of increased CD4-CTLs downstream of loss of Aiolos?

2. The authors show increased Tregs (Ext Fig 1E). In light of the role of Aiolos inhibiting CD25, does Aiolos also control Treg formation?

3. How does Aiolos alter chromatin accessibility? Is there increased H3K4me3 or H3K27Ac? Given the use of an in vitro system and the authors ability to perform ChIP, this is addressable and would improve the impact with mechanism of how Aiolos alters gene transcription.

4. NP specific tetramer staining should be shown by flow

5. When is Aiolos upregulated? At what point in T cell differentiation is it required for expression? Tfh cells are thought to form early, and if Aiolos is upstream of Bcl6, it suggests Aiolos is required early.

6. If Aiolos is upregulated and not expressed in naïve cells, what drives its expression?

7. For all in vivo experiments, cell numbers should be shown as well as percentages.

8. Analysis of ATACseq data is limited to a few genes. Are there global changes in chromatin accessibility?

AUTHORS' SUMMARY: We sincerely appreciate the thorough feedback from the Reviewers and have made substantial efforts to address their concerns and suggestions. Major concerns included the CD4⁺ T cell-intrinsic nature of our findings originally obtained from germline Aiolos KO mice, as well as a desire to see more in-depth analyses of our RNA- and ATAC-seq data. We now present a substantially revised manuscript including newly generated data, analyses, and significant revisions to the manuscript text. Specifically:

1. We now include use of an adoptive transfer system to identify CD4⁺ T cell-intrinsic roles for Aiolos in regulating CD4⁺ T cell programming events. Importantly, findings using this system support a CD4⁺ T cell-intrinsic role for Aiolos in repressing CD4-CTL responses during influenza virus infection.
2. We have substantially expanded our ATAC-seq analyses by both acquiring additional samples and performing genome-wide analyses to 1) identify statistically significant alterations in chromatin accessibility between wildtype and Aiolos-deficient T_{FH}- and T_{H1}-polarized cells, and 2) perform motif analyses to evaluate DNA binding motifs enriched at sites of statistically significant increases or decreases in accessibility. These analyses have revealed global alterations in chromatin accessibility in the absence of Aiolos in both T_{H1}- and T_{FH}-polarized cells and have identified regions of significantly decreased or increased accessibility (consistent with transcript analyses) at key T_{FH} (*Zfp831*, *Tox*, *Cd40lg*) and CD4-CTL (*Prdm1*, *Eomes*, *Ifng*, *Prf1*, *Gzmb*, *Il2ra*, *I2rb*) associated loci, respectively. Further, enriched motifs identified at significantly differentially accessible regions include STAT3, STAT5, Blimp-1, and T-box transcription factors. We make the further observation that the STAT5 motif contains the core IkZF DNA binding motif GGGAA, suggesting that Aiolos binding may play a role in antagonizing the activity of STAT5. This possibility is consistent with STAT5 ChIP data presented in Figure 8. All ATAC-seq data have been deposited in GEO, and Reviewer tokens are available at request to view the data.
3. We have expanded our ChIP qPCR analyses to assess an increased number of regions displaying significant differences in chromatin accessibility (both increased and decreased) in the absence of Aiolos. This has led to the identification of novel Aiolos target genes, as well as sites of increased STAT5 enrichment in the absence of Aiolos. Further, we also now analyze alterations in H3K27Ac enrichment at these same sites, to further describe changes to chromatin structure in the absence of Aiolos at T_{FH} and CD4-CTL associated genes. Importantly, these new findings support our original conclusion that Aiolos is a reciprocal regulator of T_{FH} and CD4-CTL programming.
4. We have substantially altered language to ensure 1) both clarity and accuracy of our findings and methods, and 2) that related prior work and implications of our current findings are accurately represented throughout the text.

Additional details of our revisions are presented in the point-by-point response below.

REVIEWER 1

Summary

Read, Jones and colleagues analyze Aiolos as a regulator of Tfh cell and CD4-CTL responses upon murine flu infection and during in vitro polarization. They find that Aiolos is necessary for Tfh cell development and antibody production in vivo, and in its absence, CD4 T cells assume a CTL fate. They further demonstrate that Aiolos positively regulates a Tfh-cell gene program, including that of Zfp831, TCF-1 and Bcl6, while repressing the expression of IL2Ra and IL-2/STAT5 signaling necessary for expression of the CTL gene program. They conclude that Aiolos serves as a central mediator of Tfh vs CTL development.

Significance

Previous work by the authors demonstrated that Aiolos (Ikzf3) can regulate the expression of Bcl6 in Tfh-like cells, suggesting it promotes their differentiation (Read, J Immunol 2017). The current paper extends this observation, demonstrating that Aiolos positively regulates Tfh cell formation in vivo and in vitro, while reciprocally regulating that of CD4 CTLs, the development of which has not been dissected as done here. These findings are strengths of the paper, alongside dissection of the gene regulatory pathways necessary for the observed phenotypic changes. That said, greater dissection of the phenotypic effects of T-cell intrinsic Aiolos regulation would strengthen the authors' conclusions. While Aiolos target gene expression is investigated, its direct regulatory relationships also could be explored in more depth.

We appreciate that the Reviewer found several strengths in our study, but also acknowledge the concerns raised by the Reviewer. We have now obtained new data that strengthen our original conclusions, notably that adoptively transferred Aiolos-KO cells upregulated CD4-CTL-like features in the lungs of influenza infected recipient mice (**Fig. 6, Extended Data Fig. 10A-D**). We also provide new ATAC-seq and ChIP data that yield further insights into the mechanisms involved (**Figs. 3, 7**).

1. Aiolos deficiency results in disrupted Tfh cell differentiation and antibody production as assessed in Irf3 germline deficient mice. While the in vitro experiments suggest this effect is CD4 T cell intrinsic, this is not the case for the in vivo effect. Rather, it could be confirmed using adoptive transfers, bone marrow chimeras, and/or CRISPR gene targeting with adoptive transfer. Such experiments would also better determine Aiolos's functional role in Tfh cells. While the reduced antibody response in germline deficiency is assumed to be secondary to reduced numbers of Tfh cells, their function (trafficking, cytokine production, CD40L expression and so forth) is not tested. Numbers of cells in addition to percentages should be shown throughout. Moreover, while flow data suggest Aiolos is necessary for GC-Tfh cell development, this is an indirect assessment, which would be better determined by GC staining, ideally including for Aiolos. While deficiency in the latter leads to reduction in Ab production 8 days following flu challenge, this time point largely analyzes early PC formation, not GC output, with the latter not determined.

Such experiments would not only rule out a non-T cell intrinsic effect but would eliminate the possibility the phenotype observed is secondary to manipulation of Tfr cells, reduced in the absence of Aiolos. To this point, the authors' reasoning that the observed 'expansion' of regulatory cells rules out a Tfh cell extrinsic effect directly contradicts their observation that Tfr cells are reduced in the germline mutants, an effect which of course can lead to reduced Tfh cell numbers and output. The only way to resolve this point is to do an experiment separating the effects of Aiolos in Tfh vs. Tfr cells.

The Reviewer makes several outstanding points. Below we have broken each individual point down (1a-1d) for ease of review and clarity of response.

1.a. Aiolos deficiency results in disrupted Tfh cell differentiation and antibody production as assessed in Irf3 germline deficient mice. While the in vitro experiments suggest this effect is CD4 T cell intrinsic, this is not the case for the in vivo effect. Rather, it could be confirmed using adoptive transfers, bone marrow chimeras, and/or CRISPR gene targeting with adoptive transfer. Such experiments would also better determine Aiolos's functional role in Tfh cells.

We appreciate the Reviewer's feedback and agreed that our findings would be strengthened by inclusion of data defining the CD4⁺ T cell-intrinsic role for Aiolos in regulating T_{FH} and CD4-CTL programming. To this end, we performed adoptive transfer studies and, in line with a CD4⁺ T cell-intrinsic role for Aiolos in regulating CD4-CTL responses, found that transferred populations in the lung exhibited increased Eomes expression, as well as augmented production of IFN- γ and the cytotoxic molecules granzyme B and perforin upon antigen stimulation (**Fig. 6 and Extended Data Fig. 10B-D**).

As the Reviewer notes, CD4⁺ T cell-intrinsic roles for Aiolos in regulating both T_{FH} and CD4-CTL programming are indeed supported by our *in vitro* findings. With specific regard to T_{FH} programming, these now include expanded analyses of ATAC- and RNA-seq data, which indicate 1) that the T_{FH} gene program is significantly disrupted in the absence of Aiolos, including reduced expression of numerous critical transcriptional regulators (*Bcl6*, *Zfp831*, *Tcf7*, *Tox*) (**Fig. 2A-B**), 2) that there are global alterations to the chromatin landscape in Aiolos-deficient cells, including significant reductions in accessibility at T_{FH} genes (*Zfp831*, *Tox*, *Cd40lg*), and enhanced accessibility at CD4-CTL-associated genes (**Fig. 3A**), and 3) that *Zfp831*, which exhibits both loss of accessibility and reduced transcript expression, is a direct target of Aiolos in this setting (**Fig. 3B-D**). Together, these data support a role for Aiolos in driving T_{FH} programming in a cell-intrinsic manner.

While we performed analyses of T_{FH} populations following adoptive transfer experiments discussed above, results from these experiments were more complex than initially anticipated:

First, our germline knockout system indicated that in addition to disrupted T_{FH} generation, loss of Aiolos resulted in significant defects in regulatory T cell populations, including both T_{REG} and T_{FR} cells (**Extended Data Fig. 3D-F**)—both of which could contribute to the observed T_{FH} deficiency. These findings are important, as they indicate for the first time that Aiolos may function to regulate T_{FR} populations. As our data both in this

manuscript and a prior study indicate that Aiolos is important for the regulation of Bcl-6, which is also expressed by T_{FR} cells, it is interesting to speculate that this may represent a shared regulatory mechanism between T_{FH} and T_{FR} populations. In contrast, during adoptive transfer, both T_{REG} and T_{FR} populations are present at normal levels (data which we now include in **Extended Data Fig. 10F-G**). This may explain, in part, why we did not observe a defect in T_{FH} populations in the lung-draining lymph node (DLN) in this experimental setting (shown in **Extended Data Fig. 10E**).

Second, while data in the lungs recapitulated our findings from Aiolos-deficient animals (i.e. that loss of Aiolos results in augmented, Eomes-dependent CD4-CTL responses (**Fig. 6** and **Extended Data Fig. 10A-D**), we did not observe the same phenotype in adoptively transferred populations in the DLN (where T_{FH} populations were also analyzed; **Extended Data Fig. 10H-I**). This suggests that either use of TCR-transgenic cells, or increased numbers of antigen-specific cells utilized in adoptive transfer, may contribute to alterations in the DLN T_{FH} and CD4-CTL responses, as has previously been reported (Olson *et al. Immunol Cell Biol.* 2016).

The above findings have now been compiled and are included in the revised version of the manuscript. Thus, while we acknowledge that we cannot rule out the potential role of B cell alterations (due to disrupted Aiolos) in the germline knockout setting, the above observations suggest that T_{FH} regulation in these settings may be more nuanced. Still, our *in vitro* findings are strongly supportive of a CD4 cell-intrinsic role for Aiolos in positively regulating the T_{FH} gene program. These considerations are detailed in the newest version of the Discussion.

1.b. While the reduced antibody response in germline deficiency is assumed to be secondary to reduced numbers of Tfh cells, their function (trafficking, cytokine production, CD40L expression and so forth) is not tested.

We appreciate this feedback from the Reviewer. Our updated analyses suggest that loss of Aiolos results both in reduced *Cd40lg* transcript expression and significantly reduced accessibility at several *Cd40lg* regulatory regions shown in (**Fig. 2A-B, Fig. 3A, Extended Data Fig. 6A**). Further, our *in vivo* findings indicate that *Cxcr5* expression is reduced in Aiolos-deficient CD4⁺ T cell populations, which could have marked effects on the ability of Aiolos-deficient cells to localize to the follicle and interact with B cells, relative to their WT counterparts (**Fig. 1B,C**). While experiments to more precisely assess T_{FH} cell trafficking and germinal center responses have been delayed due to technical hurdles, we are currently pursuing these analyses in efforts to more thoroughly understand the role of Aiolos in T_{FH} differentiation and associated functional responses.

1.c. Numbers of cells in addition to percentages should be shown throughout.

We agree with the Reviewer and have now included the numbers of evaluated influenza nucleoprotein (NP)-specific (or CD45.2⁺ donor T cell, or CD45.2⁻ recipient) populations in each tissue examined (**Figs. 1D, 5E, 6G and Extended Data Fig. 3C, 9C, and 10E-F**). Importantly, we do not observe substantial differences in the numbers of WT and Aiolos-deficient antigen-specific cells with the exception of the lungs in our germline Aiolos KO studies (**Extended Data Fig. 9C**). These data are now included and discussed in the revised manuscript.

1.d. Moreover, while flow data suggest Aiolos is necessary for GC-Tfh cell development, this is an indirect assessment, which would be better determined by GC staining, ideally including for Aiolos. While deficiency in the latter leads to reduction in Ab production 8 days following flu challenge, this time point largely analyzes early PC formation, not GC output, with the latter not determined.

Such experiments would not only rule out a non-T cell intrinsic effect but would eliminate the possibility the phenotype observed is secondary to manipulation of Tfr cells, reduced in the absence of Aiolos. To this point, the authors' reasoning that the observed 'expansion' of regulatory cells rules out a Tfh cell extrinsic effect directly contradicts their observation that Tfr cells are reduced in the germline mutants, an effect which of course can lead to reduced Tfh cell numbers and output. The only way to resolve this point is to do an experiment separating the effects of Aiolos in Tfh vs. Tfr cells.

We appreciate this feedback from the Reviewer and agree with the above points. We have now both expanded our analyses and adjusted language discussing interpretation of the data to more accurately reflect our findings. We have not yet examined later timepoints for GC output due to technical hurdles, but do plan to pursue these studies moving forward. However, we do now present data demonstrating that Bcl-6 protein is significantly reduced in Aiolos-deficient cells at an earlier timepoint during influenza virus infection (6 d.p.i.), hinting at an early T_{FH} programming defect (which we are continuing to explore; **Extended Data Fig. 3B**). We

acknowledge that disruptions to both T_{FR} and T_{REG} populations may impact the observed T_{FH} phenotype (shown in **Extended Data Figs. 3D-F**) and have modified language in both the Results and Discussion of the revised manuscript to address this.

2. To verify Aiolos targets, the authors used ATAC-seq combined with ChIP-qPCR analysis of WT and mutant cells, finding enrichment in the indicated regions of Tfh-related genes or Th1-associated genes. While compelling, it is unclear which genes are direct Aiolos targets. Also, what is the binding motif for Ikzf3? It is not clear from the data if the binding motif is found in the binding site of different Tfh associated genes.

Our combined ATAC-seq and ChIP-qPCR analyses have thus far identified *Bcl6*, *Zfp831*, and *Il2ra* as direct Aiolos gene targets in T_{FH} cells (**Figs. 3A-D, Extended Data Figs. 6C-D, 13D**). We hypothesize that additional Aiolos targets exist and are critical in regulating both T_{FH} and CD4-CTL gene profiles, which we plan to explore on a genome-wide basis in future studies. With regard to the *Ikzf3* binding motif (GGGAA), we find this motif proximal to enrichment sites of Aiolos. Further, we also find that regions of significantly altered chromatin accessibility in the absence of Aiolos are enriched for STAT5 sites which, notably, contain the core IkZF binding motif (**Figs. 3F-G, 7G**). Thus, our data are suggestive of the exciting possibility that Aiolos may function to compete with and antagonize STAT5 activity. These possibilities are detailed in the substantially revised Discussion. Together, our current findings that key T_{FH} transcriptional regulators and *Il2ra* are bound by Aiolos represents a significant conceptual advance in our understanding of the role Aiolos in CD4⁺ T cell programming events, as well as T_{FH} and CD4-CTL biology.

3. Tcf1 acts as a master transcriptional regulator of T cells, with expression in naïve T cells and maintenance in Tfh cells. Is Tcf1 reduced in naïve cells or only specifically in Tfh cells in Aiolos-deficient mice? Is downregulation of Tcf1 only observed under TFH-like polarizing culture conditions? It would be good to know whether this is specific for Tfh cells. Is the phenotype of Aiolos knockout mice like that of Tcf1 knockout mice, and can overexpression of Tcf1 or other targets rescue the Aiolos-deficient phenotype? Is regulation of Tcf1 by Aiolos universal or specific to Tfh cells?

New transcript data from wildtype versus Aiolos-deficient naïve CD4⁺ T cells demonstrate that the effect of Aiolos deficiency on TCF-1 expression does not extend to naïve T cells (**Extended Data Fig. 4C**). We hypothesize that the activity of Aiolos may be altered in mature effector populations vs. early T cell precursors due to altered partner proteins. For example, our earlier work demonstrates that Aiolos interacts with STAT3 downstream of cytokine signals received during effector differentiation (Read *et al. J Immunol.* 2017). Thus, interaction with STAT3 may alter the effect of Aiolos on gene programs in effector cells versus developing T cells. An intriguing aspect of our data is that *Tcf7* (*Tcf-1*) expression is decreased not only in T_{FH} cells, but also in Aiolos-deficient T_{H1} cells that also display an increased cytotoxic gene signature (**Figs. 2A, 4G**). This is consistent with the findings from Donnarumma *et al. Cell Reports* 2016, demonstrating that TCF-1 expression can oppose the acquisition of cytotoxic features by CD4 T cells.

Minor Points (as indicated by Reviewer 1)

1. Il2ra is increased in Ikzf3-/- cells; however, the effect is relatively small (~1.5-fold), suggesting IL-2ra may not be a primary Aiolos target.

We appreciate the comment from the Reviewer and understand the concern over the small transcriptional effect. However, we now show that in the absence of Aiolos, CD25 and CD122 expression are augmented at the transcript level *in vitro* (**Figs. 2B, 4G, Extended Data Fig. 13A**), and at the protein level *in vivo* (**Figs. 5F, 6H-I, Extended Data Figs. 9F, 10A**). Conversely, we now show that overexpression of Aiolos represses expression of both *Il2ra* and *Il2rb* at the transcript level (**Extended Data Fig. 13B**). Functionally, we observe that Aiolos deficiency results in increased STAT5 tyrosine phosphorylation-mediated activation in both T_{FH} and T_{H1}-polarized cells (**Fig. 8A, Extended Data Fig. 13C**). Our expanded ATAC-seq data also support a role for Aiolos in regulating CD25 (and CD122) expression (**Figs. 3A, 7A-C, and Extended Data Fig. 12**).

Despite relatively modest transcript changes, these findings strongly support a phenotypic and functional effect for Aiolos-dependent alterations at the *Il2ra* and *Il2rb* loci.

2. The paper dissects the “Zfp831/Tcf-1/Bcl6 axis”, yet this is not shown in the graphic model (Extended Data Figure 7)

We thank the Reviewer for this observation and have made necessary adjustments to the graphic model (**Extended Data Fig. 14**).

3. The authors show the Tfh associated gene expression in in vitro cells cultured under Tfh-like polarizing conditions. How about the expression of Tcf-1, Bcl-6, and Zfp831 in primary Tfh cells from wild-type and Aiolos knockout mice?

We agree with the Reviewer that this is an important point. While we have not performed a comprehensive analysis on all T_{FH} transcriptional regulators (analysis of Zfp831 is hindered by the lack of a commercially available flow cytometry antibody), our updated analysis of Bcl-6 protein expression during influenza virus infection (6 days post infection) reveals a statistically significant reduction in Bcl-6 protein in the absence of Aiolos (**Extended Data Fig. 3B**). These data are suggestive of an early T_{FH} defect. We look forward to examining the expression of additional key T_{FH} transcription factors at the protein level in future work.

4. The Stat5 ChIP-seq peak is missing in the IGV figure (Extended Data Figure 6).

IGV figures have been substantially expanded, and each track (where applicable), includes STAT5 ChIP-seq tracks (**Fig. 7B-F, Extended Data Fig. 12**).

REVIEWER 2

In this manuscript Read KA and colleagues investigated the role of the transcription factor Aiolos in Tfh cells. In a previous study from the same group (Read KA J Immunol 2017), the authors found that Aiolos expression was increased in antigen-specific Tfh cells after influenza infection, compared to antigen-specific effector T cells. This new study now aims at extending on this finding, by showing that in the absence of Aiolos, Tfh cell differentiation in response to influenza antigens was impaired, leading to reduced antibody response. On the other hand, T cells lacking Aiolos that were cultured under Tfh conditions acquired some features of a cytotoxic-like program, including perforin expression. Aiolos deficiency also resulted in increased CD25 expression.

While some observations warrant further investigation (for example, the role of Aiolos in CD4-CTLs, or in T cell responses to viral infections), other findings appear to be more incremental and preliminary, and in the absence of more thorough and global analyses I remain uncertain whether some of the conclusions are indeed adequately supported by experimental evidence.

We thank the Reviewer for their feedback and agree that more thorough and global analyses were required to support our findings. These include:

1. Global ATAC-seq analyses for WT vs. Aiolos-deficient T_{FH}⁻ and T_H1-polarized cells (**Extended Data Figs. 5, 11**).
2. Analysis of statistically significant differences in accessibility throughout the genomes of WT vs. Aiolos-deficient cells (**Figs. 3A, 7A**).
3. Motif analyses within statistically significant regions of altered chromatin accessibility between WT and Aiolos-deficient cells (**Figs. 3F-G, 7G-H**).
4. Expanded analyses of full gene loci of interest (**Extended Data Figs. 6, 12**).
5. Expanded RNA-seq analyses to assess 1) unique and overlapping DEGs between T_{FH}⁻ and T_H1-polarized cells (**Extended Data Fig. 8A-B**) and 2) alterations in well-established activation markers T_{FH}⁻ and T_H1-polarized cells (**Extended Data Fig. 8C-D**).

These have been incorporated into the substantially revised manuscript and are further detailed in the Authors' Summary and throughout this document. Also, please note that Point 1 has been subdivided based on each study that the Reviewer refers to.

1. Currently this study is not adequately put into the context of previous relevant work on Aiolos published by other labs. Most notably:

- Quintana et al (Nature Immunology 2012) showed that Aiolos controlled the differentiation and function of Th17 cells, and that Aiolos was most highly expressed in this subset. In Figure 2a, how does expression of Aiolos in Tfh cells compare to Th17 cells? This information would provide more general insights on its role in T cell differentiation.

We now provide these data and find that Aiolos expression is elevated in both T_{FH} and T_H17 cells relative to other T helper cell populations (**Extended Data Fig. 1A-B**). Further, we have adjusted language in both the

Results and Discussion sections to more thoroughly place our findings into the context of (or, rather, compare and contrast with) previous work in T_H17 populations. It is also important to note that our analyses are performed in the context of infection, rather than autoimmune settings (as with prior T_H17 cell work), which is an important distinction.

- Kuehn *et al* (*J Exp Med* 2021) showed that a mutation in AIOLOS identified in human patients was associated to T and B cell abnormalities (including impaired T_{fh} cell differentiation), recapitulated in a mouse model. In Figure 1, can the authors rule out that reduced antibody production is not due to an intrinsic B cell defect?

As discussed in response to Reviewer 1, while our *in vitro* data are strongly supportive of a CD4 cell-intrinsic role for Aiolos in regulating T_{FH} programming, we cannot rule out some contribution of B cells or regulatory T cell populations to the observed reduction in T_{FH} cell populations and antibody production. We also now include a discussion of the above work, though it is important to note that the AIOLOS mutation in question resulted in a dominant negative form of Aiolos, rather than its deficiency, as explored in the current manuscript.

- Wang JH (*Immunity* 1998) showed increased proliferation of Aiolos-deficient T cells in response to TCR activation. Can the authors rule out that some of the differences they observe in the current study are not due differences in T cell activation?

We appreciated this feedback from the Reviewer and acknowledge that Aiolos has previously been implicated in regulation of T cell activation. To address whether this was also the case in our analyses, we analyzed expression of numerous cell activation markers by Aiolos-deficient T_H1 and T_{FH}-polarized cells (**Extended Data Fig. 8C-D**). Our findings do not indicate that a global change in cell activation is responsible for the observed phenotypes. Furthermore, we do not observe a significant difference in the frequency or number of antigen-specific populations in either the DLN or lungs following adoptive transfer of wildtype or Aiolos-deficient populations, suggesting again that global T cell activation/proliferation differences are not driving the observed differences in T_{FH} and CTL populations (**Fig. 6G and Extended Data Fig. 10E**).

2. Figure 3: since the authors performed ATAC-seq on T_{fh} cells, they should assess if a more comprehensive analysis (rather than just looking at a few pre-selected genes) provides more information about the regulatory roles of Aiolos in T_{fh} cells. In general, I don't think that based on a few selected snapshots the authors can draw the conclusion that "chromatin accessibility was negatively impacted by the absence of Aiolos". There might be many more regions where chromatin accessibility is positively impacted. And the slight reduction shown could simply reflect experimental variability. Also, how do these differences compare to differences in accessibility at established Aiolos targets, like *Il2*?

We appreciate this feedback from the Reviewer and agree with this assessment. As noted above, we have now performed genome-wide ATAC-seq analyses, and indeed observe global changes in chromatin accessibility (both augmented and reduced) in the absence of Aiolos (**Extended Data Figs. 5, 11**). Importantly, these included statistically significant alterations in accessibility at key target loci (**Figs. 3A, 7A**). Full loci are also presented to strengthen our observations, and regions of significantly altered accessibility have been noted (**Extended Data Figs. 6, 12**). In contrast to key T_{FH} and CD4-CTL associated genes, we do not observe differences in accessibility at the *Il2* locus in the absence of Aiolos in this setting (**Extended Data Figure 12**). We speculate that this may be due to 1) differences between autoimmune (Quintana *et al. Nat Immunol* 2012) and infection (present study) settings, and/or 2) differences in IL-2 expression in T_{FH} vs. T_H17 cells, as T_{FH} cells, unlike T_H17 cells, are known IL-2 producers (DiToro *et al. Science* 2018).

3. Still in Figure 3, a better control of specificity for the ChIP-PCR assay is represented by the *Ikzf3*^{-/-} T_{fh} cells, which should be used in comparison with the other experimental groups.

We agree with the Reviewer, and now present these data here. These findings (at *Zfp831*) demonstrate that limited background for Aiolos IP is detected in Aiolos-deficient cells.

4. The authors could establish if Aiolos indeed directly regulates the expression of some of the identified factors (*Bcl6*, *Il2ra*, etc) by using gain-of-function experiments and luciferase reporter assays.

We appreciate the suggestion from the Reviewer and have indeed shown that Aiolos augments the promoter activity of *Bcl6* in a prior publication (Read *et al.*, *J Immunol* 2017). Further, we now include Aiolos overexpression data in T_H1 cells, which indicates that overexpression of Aiolos is sufficient to repress both *Il2ra* and *Il2rb* transcript expression (**Extended Data Fig.13B**, and shown below). Conversely, we observe that overexpression of Aiolos results in a trending increase in the expression of the T_{FH} gene *Zfp831* (shown below), which is downregulated in the absence of Aiolos (**Figs. 2B, 4G**), and which we have identified as a direct Aiolos target (**Fig. 3D**).

5. The authors state (page 10) that Aiolos directly regulates CD25 expression. However, the authors only showed that the *Il2ra* transcript is expressed comparatively at higher levels (~2 times higher) in *Ikzf3*^{-/-} Tfh cells compared to wild-type. How does this translate into the dynamics of CD25 expression at different time points of T cell activation and with different strengths of stimulation? A small upregulation of CD25 expression may end up being modest and primarily linked to increased T cell activation, rather than differentiation. B cells lacking Aiolos for example exhibited an activated phenotype (Wang *JH Immunity* 1998).

We appreciate this feedback from the Reviewer. As discussed above, our data are not consistent with overall increases in CD4⁺ T cell activation (**Extended Data Fig. 8C-D**). As included in response to Reviewer 1 point 4, our combined *in vitro* and *in vivo* expression data (**Fig. 2B, 4G, 5F, 6H-I, Extended Data Figs. 9F, 10F, 13A-B**) when coupled with the observed changes in STAT5 tyrosine phosphorylation (**Fig. 8A, Extended Data Fig. 13C**), support a strong phenotypic and functional effect of Aiolos with regard to the repression of IL-2/STAT5 signaling. Furthermore, our ATAC-seq and ChIP data (**Figs. 3A, 7A-C, Extended Data Fig. 12, 13D**) support a direct role for Aiolos in regulating IL-2Ra expression.

6. Along the same line, while I appreciate that the authors show increased CD25 expression *in vivo* (Figure 7), this could also be due to lower thresholds of T cell activation in absence of Aiolos. What about the surface expression of other activation markers (both *in vivo* and *in vitro*) that are not regulated by Aiolos? Including also *Icos*, which is central to Tfh biology.

Our data demonstrate that activation markers, including *Icos*, are not universally upregulated by Aiolos-deficient cells (**Extended Data Fig. 8C-D**), as detailed in response to Reviewer 2, point 1.

7. Please also indicate where all the relevant transcripts (including *Il2ra*, *Il2rb*, *Icos*) are located in the volcano plots in Figure 2 and 5.

We agree that this information should be included; these labels have now been added to the volcano plots (**Fig. 2A, Extended Data Fig. 7D**).

8. A more in-depth comparison of the RNA-seq data in Tfh and Th1 cells could provide more information about the role of Aiolos in these cells. What is the extent of overlap of the up- and down-regulated genes in Tfh and Th1 conditions in the absence of Aiolos? Some Venn diagrams with indicated common and unique genes would already be informative. If mostly the same genes are dysregulated in these two cell subsets upon Aiolos deletion, then the main phenotype may be linked to general T cell activation rather than differentiation.

We agree that expanded analyses of RNA-seq findings would be beneficial. We have now analyzed unique and shared genes between up- and down-regulated genes in T_H1 and T_{FH} data (**Extended Data Fig. 8A-B**). These findings are now included and indeed demonstrate that while there are some similarities in the genes regulated by Aiolos expression in T_H1 and T_{FH} cells, there are also many differences.

9) Figure 6F: was *Cx3cr1* expression increased also in any of the RNA-seq datasets? Excel files containing at the very least the significant differentially expressed genes and differentially accessible regions should be made available to be able to assess the consistency and quality of the data. The number of biological replicates used in RNA-seq for Tfh cells is not indicated neither in the methods nor in the figure legends. Please show PCAs of all RNA-seq and ATAC-seq experiments. Legend of Figure 3 mentions two replicates for ATAC-seq, which is insufficient to achieve robust results.

The requested data have been added as an **additional supplementary file** and now include:

1. The top 200 significantly differentially expressed genes for each cell type
2. PCA plots for all ATAC-seq and RNA seq analyses
3. Expanded biological replicates for T_H1 ATAC-seq samples (n=3), and associated analyses
4. The top 500 significantly differentially accessible regions for each cell type
5. The top 60 enriched motifs within regions of both increased and decreased accessibility for each cell type

Additionally, while we found the *Cx3cr1* data to be encouraging, we have instead chosen to focus on more established markers of CD4-CTLs including NKG2A/C/E (Workman et al. *Plos One*, 2014 and Marshall et al. *J Immunol*, 2017) in the revised manuscript (**Figs. 4G, 6E, 7A and Extended Data Fig. 12**).

10) Based on the CD25 data presented in this manuscript, and in the absence of other information (that is, expression of other activation markers not regulated by Aiolos, functional effects of IL-2 on cell proliferation and so on), the conclusion that “Aiolos represses IL-2 responsiveness” (page 12) is both premature and preliminary, since no direct evidences are presented.

We agreed with the Reviewer that expanded analyses would strengthen our findings regarding the role of Aiolos in regulating IL-2 responsiveness. As discussed above, we now include:

1. Global ATAC-seq analyses, which identify regions of both *Il2ra* and *Il2rb* as statistically significant in terms of differential (enhanced) accessibility in the absence of Aiolos (**Figs. 3A, 7A-C, Extended Data Figs. 5, 11**).
2. Assessment of numerous activation markers, which indicate that Aiolos deficiency does not globally impact T cell activation (**Extended Data Fig. 8C-D**).
3. Analysis of CD25 and CD122 surface expression during influenza virus infection (**Figs. 5F, 6H-I, Extended Data Figs. 9F, 10A**).
4. Analysis of STAT5 tyrosine phosphorylation-mediated activation in the absence of Aiolos (**Fig. 8A, Extended Data Fig. 13C**).
5. Broad analysis of STAT5 enrichment at CD4-CTL gene targets (including *Il2ra*) in the absence of Aiolos (**Fig. 7B-F, Extended Data Fig. 12**).

Importantly, we do not observe alterations in accessibility at the *Il2* locus in the absence of Aiolos (**Extended Data Fig. 12**). Together, these data are strongly supportive of a role for Aiolos in regulating IL-2 responsiveness and downstream STAT5 activation/activities.

11) I am uncertain what conclusions can be drawn from Figure 8: is the increased STAT5 phosphorylation actually due to IL-2 stimulation? Is it significant? Does it occur also in Aiolos-deficient Tfh cells? Does it affect T cell responses to IL-2?

Our data in **Fig. 8A and Extended Data Fig. 13C** demonstrate that STAT5 phosphorylation is augmented in the absence of Aiolos. These data when combined with increased IL-2R expression and the pathway analysis in both Aiolos-deficient T_H1 and T_{FH} cells (**Fig. 2C, 4I**) support the assertion that IL-2 signaling (but not production, as shown in **Figure 5G**) is increased in the absence of Aiolos. Furthermore, we find enhanced STAT5 enrichment at CTL genes in the absence of Aiolos (**Fig. 8B-D**). Finally, using HOMER motif analysis, we find that the STAT5 DNA-binding motif is one of the most overrepresented motifs in areas of significantly increased chromatin accessibility in the absence of Aiolos (**Figs. 3F, 7G**). We have altered the language describing our findings to be as precise as possible.

12) If Aiolos indeed suppresses CD25 expression, then its retroviral transduction in wt and Aiolos-deficient cells should reduce (or normalize, respectively) CD25 expression and any downstream functional effects.

As discussed in point 4, we now include Aiolos overexpression data in T_H1 cells, which indicates that overexpression of Aiolos is sufficient to repress both *Ii2ra* and *Ii2rb* transcript expression (**Extended Data Fig. 13B**).

13) The observation that T cells lacking Aiolos express granzyme and perforin is interesting: do they also acquire killing capacity in vitro?

In our revised manuscript, at 8 days post-influenza infection in the adoptive transfer model, we generate whole-tissue homogenates containing donor CD45.2-OT-II WT or Aiolos-deficient cells, which are stimulated *ex vivo* with OVA peptide. We find that in response to antigen stimulation, Aiolos-deficient cells produce increased IFN- γ , perforin, and granzyme b, relative to their WT counterparts, suggestive of increased killing capacity (**Fig. 6F and Extended Data Fig. 10B-D**).

14) It is unclear from the figure legend if the human data shown in Extended Figure 1 show 3 independent donors. I also believe that the graph should show SD and not SEM and paired, rather than unpaired t-test. The authors may want to double-check their statistics throughout the manuscript.

We appreciate this feedback from the Reviewer and have made efforts to review all statistics in the manuscript; these have been altered as necessary and are appropriate for individual experiments. We have further corrected language in the indicated figure legend to indicate individual donors (now **Extended Data Fig. 2B-E**).

REVIEWER 3

In the manuscript by Read et al, the authors explore the role of Aiolos in Tfh/CD4 CTL differentiation. In line with their previous publication (Read et al, J. Immunol 2017) the authors find that Aiolos supports Bcl6 and subsequent Tfh differentiation, in this study by using Aiolos (Ikzf3) deficient mice. The loss of Tfh cells leads to a subsequent increase in CD4 T cells that have cytotoxic capabilities, including perforin and Granzyme B, as well as IFN γ . This is attributed to alterations in the chromatin accessibility and binding of Aiolos to Tfh genes, and in contrast, a repressive effect on CD25 leading to increased STAT5 signaling and downstream genes associated with CTL function. While the data are straightforward and good quality, the findings are not surprising in light of prior work from this group and others (Quintana et al, Nat. Immunol. 2012) that Aiolos plays a key role in T cell differentiation by promoting Bcl6, Tfh and inhibiting the IL-2 pathway. There are also several missed opportunities that would bolster the impact of these findings, as well as key deficiencies in experimental design that limit the impact.

We appreciate that the Reviewer found the data to be of good quality. We also appreciate the thorough critique and suggestions for improving the study. We have now obtained new data that both expand the scope of our study and strengthen our original conclusions.

1. Most of the data presented are in vitro Th1 and Tfh-like cells. In vivo data with influenza have limited attempts to address functional impact beyond one experiment for antibody formation. In LN, GCs should be assessed if Tfh are decreased. Also missed opportunity to look at CD4-CTLs in the lung tissue. Brown et al (J. Virol. 2012) demonstrated a protective role for CD4-CTLs in influenza infection. Does loss of Aiolos lead to increased viral clearance in the lungs, or perhaps increased immunopathology? What are the functional consequences of increased CD4-CTLs downstream of loss of Aiolos?

We appreciate the feedback from the Reviewer and now present data from the lungs (in an adoptive transfer setting) showing that cells with augmented cytotoxic features are indeed significantly increased in Aiolos-deficient transferred populations relative to their WT counterparts (**Fig. 6** and **Extended Data Fig. 10B-D**). Further, data demonstrating reduced weight loss in Aiolos-deficient animals suggest that they may be resistant to influenza infection (presented below).

Aiolos deficient mice are resistant to weight loss following influenza infection. WT and *Ikzf3*^{-/-} mice were infected with PR8-influenza virus and weights were recorded every day. Weights are presented as percent loss compared to initial weight when infected. Data are representative of 3 independent experiments. (n = 9 ± s.e.m; *P<0.05, ***P < 0.001, ****P<0.0001; multiple unpaired Student's t-test)

2. The authors show increased Tregs (Ext Fig 1E). In light of the role of Aiolos inhibiting CD25, does Aiolos also control Treg formation?

We appreciate this question from the Reviewer and find (as discussed in detail in response to Reviewer 1), that Foxp3⁺ T_{REG} cells are significantly elevated in the absence of Aiolos (**Extended Data Fig. 3D-E**). We are currently exploring this relationship more thoroughly in a separate study. It is important to note that in our adoptive transfer model we see normal T_{REG} numbers as well as a strong increase in CD4-CTL programs in the lungs (as detailed above).

3. How does Aiolos alter chromatin accessibility? Is there increased H3K4me3 or H3K27Ac? Given the use of an in vitro system and the authors ability to perform ChIP, this is addressable and would improve the impact with mechanism of how Aiolos alters gene transcription.

We appreciate these questions from the Reviewer and now find that our data are consistent with two mechanisms:

1. At key T_{FH} gene loci (at sites of observed Aiolos enrichment in WT cells), loss of Aiolos correlates with a significant decrease in H3K27Ac (**Fig. 3E and Extended Data Fig. 6E**).
2. In contrast, at genes associated with the CD4-CTL differentiation program, we observe not only increased STAT5 enrichment, but also increased H3K27Ac in the absence of Aiolos (**Fig. 8E**).

Collectively, these findings suggest that Aiolos positively impacts chromatin accessibility at T_{FH} genes, and negatively impacts chromatin accessibility at CTL genes via suppression of IL-2 signaling and STAT5 enrichment.

4. NP specific tetramer staining should be shown by flow.

We apologize for this omission. These data are now included in our gating strategy (**Extended Data Fig. 3A**).

5. When is Aiolos upregulated? At what point in T cell differentiation is it required for expression? Tfh cells are thought to form early, and if Aiolos is upstream of Bcl6, it suggests Aiolos is required early.

New data demonstrate that Aiolos expression is upregulated 24h after cell activation and correlates positively with expression of the key T_{FH} transcriptional regulator Bcl-6 (**Extended Data Fig. 1E-F**). Importantly, this upregulation of Aiolos was not observed in T_{H0} cells (cultured on stimulation, but under non-polarizing conditions), suggesting that Aiolos expression is induced by cytokine, not stimulation, signals.

6. If Aiolos is upregulated and not expressed in naïve cells, what drives its expression?

A prior study has shown that Aiolos expression was regulated by STAT3 and Ahr in T_{H17} cells (Quintana *et al. Nat. Immunol.* 2012). We are currently interrogating these possibilities in T_{FH} cells (i.e. IL-6/STAT3) in a separate study. However, our findings in the current study (discussed in point 5) suggest that cytokine signals are required for the induction of Aiolos.

7. For all in vivo experiments, cell numbers should be shown as well as percentages.

As detailed in our response to Reviewer 1, we have now included the numbers of evaluated influenza nucleoprotein (NP)-specific (or CD45.2⁺ donor T cell, or CD45.2⁻ recipient) populations in each tissue examined (**Figs. 1D, 5E, 6G and Extended Data Fig. 3C, 9C, and 10E-G**).

8. Analysis of ATACseq data is limited to a few genes. Are there global changes in chromatin accessibility?

We now present these data and indeed observe global changes to chromatin accessibility in the absence of Aiolos in both T_H1 and T_{FH} cells, detailed in previous responses.

We thank the Reviewers for their time and careful attention to revising our manuscript. We hope that the Reviewers will agree that the work detailed above has significantly strengthened our manuscript and now find it acceptable for publication in *Nature Communications*.

REVIEWER COMMENTS

Reviewer #1 (Remarks to the Author):

The newly added data, including the CD4+ T cell-intrinsic analysis and expanded details of their RNA- and ATAC-seq data, address concerns and improve the quality of the manuscript. The authors might consider the following suggestions:

1) The authors should clarify why the number of DEGs (Fig. 2A) is different in the revised version compared to the first version (in this version -- up 276, down 207 -- compared to the first version -- up 366, down 316) since the same criteria, p value < 0.05 and absolute log₂ fold change of >1.0, were used for assessment.

2) In the in vitro Tfh cell differentiation, were cells analyzed by flow cytometry? If so, it would be helpful to show these data. Comparison to gene expression of in vivo derived Tfh cells might also help confirm the in vitro polarization conditions.

3) Is *Ikzf3* differentially expressed in RNA-seq of Th1 and Tfh cell populations? It is not shown in the top 200 DEGs. It would be helpful to provide the full list of the latter.

4) Comparison of Fig 3B and Extended Figure 6A is confusing with the latter showing two genes. Please clarify. It also seems that the *Zfp831* locus analyzed this time is different loci than the previous version.

5) In Extended Data Figure 11C, the volcano plot shows the top 10,000 regions of altered accessibility between Th1-polarized *Aiolos*-deficient vs. WT cells. Is the lack of significance at 0.05 due to lack of gray points in this region?

Reviewer #2 (Remarks to the Author):

In this revised manuscript, the authors addressed the majority of my previous concerns. One remaining weakness is related to the analysis of T cell activation markers. Many of these markers are modulated very early (within 24h of stimulation) and some of them are modulated at the post-transcriptional level. Therefore, assessing late (day 3) transcriptional events is insufficient to draw conclusions on this point. The authors may want to at least revise their text on this.

Other small points:

- It looks like the difference in Supplementary Figure 1B would become significant with an appropriate number of biological replicates
- Page 13: the authors state that there is no difference in the percentage of lung Tbet+ NP-specific cells when in fact it is decreased (Suppl. Figure 9a)

Reviewer #3 (Remarks to the Author):

In the revised manuscript by Read et al, the authors have extended their studies on *Aiolos* to assess its intrinsic role in the Tfh/CD4-CTL differentiation process, added more extensive mechanistic studies on the role of *Aiolos* in chromatin accessibility and altering IL-2 signaling via STAT5 to antagonize the CD4-CTL pathway. While not all of the new data fit the authors

initial hypothesis, notable with regard to the intrinsic in vivo role of Aiolos to promote Tfh cells, they have done a good job of including these results and addressing them in the discussion. I would like to see the weight loss graph included in the paper as the functional role of increased CD4-CTLs is important to include as a readout of Aiolos function in vivo. Otherwise all of my prior comments have been suitably addressed.

AUTHORS' SUMMARY: We very much appreciate the thoughtful feedback provided by the Reviewers and were pleased to note that they felt we addressed the majority of their concerns. We thank the Reviewers for providing a few additional suggestions designed to improve the clarity of the data presented and conclusions drawn from these data. We have now addressed the remaining concerns as outlined in the following point-by-point response:

Reviewer #1 (Remarks to the Author):

The newly added data, including the CD4⁺ T cell-intrinsic analysis and expanded details of their RNA- and ATAC-seq data, address concerns and improve the quality of the manuscript. The authors might consider the following suggestions:

1) The authors should clarify why the number of DEGs (Fig. 2A) is different in the revised version compared to the first version (in this version -- up 276, down 207 -- compared to the first version -- up 366, down 316) since the same criteria, p value < 0.05 and absolute \log_2 fold change of >1.0 , were used for assessment.

We appreciate this observation from the Reviewer. In the initial submission, reported differentially expressed genes (DEGs) were presented based on p value <0.05 ; this has been corrected (including all associated analyses and figures) to reflect analyses based on an adjusted p value <0.05 . As an assessment based on adjusted p value is more rigorous, this resulted in updated DEG numbers.

2) In the *in vitro* Tfh cell differentiation, were cells analyzed by flow cytometry? If so, it would be helpful to show these data. Comparison to gene expression of *in vivo* derived Tfh cells might also help confirm the *in vitro* polarization conditions.

We appreciate this point of clarification from the reviewer. *In vitro*-polarized Tfh cells were not analyzed via flow cytometry for this manuscript. However, previous studies, including work from our laboratory, suggests that CD4⁺ T cells polarized in the presence of STAT3-activating cytokines (such as IL-6, IL-21), exhibit both phenotypic and functional characteristics associated with *bona fide* Tfh populations (PMID: 30487586, PMID: 31570752, PMID: 22018472, PMID: 31732165).

3) Is *Ikzf3* differentially expressed in RNA-seq of Th1 and Tfh cell populations? It is not shown in the top 200 DEGs. It would be helpful to provide the full list of the latter.

We appreciate this observation from the Reviewer and would like to note that *Ikzf3* transcript counts are present for exons 4-8 of Aiolos-deficient (*Ikzf3*^{-/-}) samples. However: a) RNA-seq visualization via IGV (figure at right) displays a complete lack of coverage for exons 1-3 (consistent with the *Ikzf3* KO), b) qRT-PCR analysis verified lack of *Ikzf3* transcript for all KO samples (Fig. 4a, Extended Data Fig. 3a), and c) immunoblot analyses verify absence of Aiolos protein (using Active Motif 39293, raised against the C-terminal portion of Aiolos) (Fig. 8a, Extended Data Figs. 3b, 7a, 13c). The full list of DEGs can be found in the source data (for Fig. 2a, Extended Data Fig. 7d) and has also been deposited under GEO Accession # GSE203066.

4) Comparison of Fig 3B and Extended Figure 6A is confusing with the latter showing two genes. Please clarify. It also seems that the *Zfp831* locus analyzed this time is different loci than the previous version.

We appreciate this observation by the reviewer. While the location of primers utilized for ChIP analyses did not change, we updated ATAC-seq analyses to utilize the mouse genome assembly mm10 (mm9 was used for the initial submission) to ensure consistency between cell types for both accuracy and clarity. As a result, the *Zfp831* locus annotation was updated (and now overlaps with another annotation (XR_001783595.1, shown at right). While the analyzed region is still present at the same *Zfp831* enhancer region, we have elected to update the figures with expanded RefSeq Gene tracks for clarity (Fig. 3b, Extended Data Fig. 6a).

5) In Extended Data Figure 11C, the volcano plot shows the top 10,000 regions of altered accessibility between Th1-polarized Aiolos - deficient vs. WT cells. Is the lack of significance at 0.05 due to lack of gray points in this region?

Limitations in the software used to analyze this dataset precluded visual presentation of the full repertoire of differentially accessible regions (DARs). As such, these plots represent the top 10,000 most significant DARs to illustrate regions of increased and decreased accessibility. However, this is not an exhaustive representation. (Additional regions exist which are closer to $p=0.05$, but are not shown). ATAC-seq datasets have been uploaded and are available via GEO Accession #GSE203066.

Reviewer #2 (Remarks to the Author):

In this revised manuscript, the authors addressed the majority of my previous concerns. One remaining weakness is related to the analysis of T cell activation markers. Many of these markers are modulated very early (within 24h of stimulation) and some of them are modulated at the post-transcriptional level. Therefore, assessing late (day 3) transcriptional events is insufficient to draw conclusions on this point. The authors may want to at least revise their text on this.

We appreciate this point from the reviewer and have updated language in the text to more accurately represent conclusions that can be drawn from these data (highlighted in yellow in the text).

Other small points:

- It looks like the difference in Supplementary Figure 1B would become significant with an appropriate number of biological replicates

We agree with the reviewer; this may warrant further investigation into the role of Aiolos in Th17 versus Tfh populations in the future.

- Page 13: the authors state that there is no difference in the percentage of lung Tbet+ NP-specific cells when in fact it is decreased (Suppl. Figure 9a)

We appreciate this observation by the reviewer and have now corrected this mistake (highlighted in yellow in the text).

Reviewer #3 (Remarks to the Author):

In the revised manuscript by Read et al, the authors have extended their studies on Aiolos to assess its intrinsic role in the Tfh/CD4-CTL differentiation process, added more extensive mechanistic studies on the role of Aiolos in chromatin accessibility and altering IL-2 signaling via STAT5 to antagonize the CD4-CTL pathway. While not all of the new data fit the authors initial hypothesis, notable with regard to the intrinsic in vivo role of Aiolos to promote Tfh cells, they have done a good job of including these results and addressing them in the discussion.

I would like to see the weight loss graph included in the paper as the functional role of increased CD4-CTLs is important to include as a readout of Aiolos function in vivo. Otherwise all of my prior comments have been suitably addressed.

We appreciate this point from the reviewer and now include weight loss data in the manuscript in **Extended Data Fig. 9a**.

We thank the Reviewers for their time and careful attention to our manuscript. We hope that the Reviewers will agree that the work detailed above has strengthened our manuscript and now find it acceptable for publication in *Nature Communications*.

REVIEWERS' COMMENTS

Reviewer #1 (Remarks to the Author):

The authors have addressed my concerns. No further comments.

Reviewer #3 (Remarks to the Author):

The authors have addressed all my concerns.

AUTHORS' SUMMARY: We again very much appreciate the thoughtful feedback provided by the Reviewers and were pleased to note that the Reviewers feel we have addressed their concerns.

Reviewer #1 (Remarks to the Author):

The authors have addressed my concerns. No further comments.

Reviewer #3 (Remarks to the Author):

The authors have addressed all my concerns.

We thank the Reviewers for their time and careful attention to our manuscript.